# A global meta-analysis on the effects of organic and inorganic fertilization on grasslands and croplands

Ting-Shuai Shi[1], Scott L. Collins [2], Kailiang Yu [3], Josep Peñuelas [4,5], Jordi Sardans [4,5], Hailing Li[1] & Jian-Sheng Ye [1] ✉

A central role for nature-based solution is to identify optimal management practices to address environmental challenges, including carbon sequestration and biodiversity conservation. Inorganic fertilization increases plant aboveground biomass but often causes a tradeoff with plant diversity loss. It remains unclear, however, whether organic fertilization, as a potential nature-based solution, could alter this tradeoff by increasing aboveground biomass without plant diversity loss. Here we compile data from 537 experiments on organic and inorganic fertilization across grasslands and croplands worldwide to evaluate the responses of aboveground biomass, plant diversity, and soil organic carbon (SOC). Both organic and inorganic fertilization increase aboveground biomass by 56% and 42% relative to ambient, respectively. However, only inorganic fertilization decreases plant diversity, while organic fertilization increases plant diversity in grasslands with greater soil water content. Moreover, organic fertilization increases SOC in grasslands by 19% and 15% relative to ambient and inorganic fertilization, respectively. The positive effect of organic fertilization on SOC increases with increasing mean annual temperature in grasslands, a pattern not observed in croplands. Collectively, our findings highlight organic fertilization as a potential nature-based solution that can increase two ecosystem services of grasslands, forage production, and soil carbon storage, without a tradeoff in plant diversity loss.

Global environmental change, such as increased atmospheric $CO_2$ and nitrogen enrichment, continues to threaten the ability of ecosystems to provide critical functions and services[1]. These challenges have motivated the development of nature-based solutions (e.g., organic fertilization and avoiding over-application of inorganic fertilizers) to mitigate the consequences of global environmental change through biodiversity conservation and increased carbon sequestration[2,3]. Yet, designing and implementing nature-based solutions that can address multiple sustainability goals require

research targeted towards specific ecosystems and conservation objectives[4].

Inorganic fertilizers have been widely applied to increase plant productivity in grassland and cropland ecosystems[5,6]. However, in grasslands, this management practice often alters plant community composition and diversity[6,7]. Indeed, most previous studies that have evaluated the effects of inorganic nitrogen[6,8] have found a general tradeoff occurs between increased aboveground biomass and lower plant diversity, with larger plant diversity losses as the amount and

[1]State Key Laboratory of Herbage Improvement and Grassland Agro-ecosystems, College of Ecology, Lanzhou University, Lanzhou 730000, China. [2]Department of Biology, University of New Mexico, Albuquerque, NM 87131, USA. [3]High Meadows Environmental Institute, Princeton University, Princeton, NJ, USA. [4]CSIC, Global Ecology Unit CREAF-CSIC-UAB, Barcelona 08193, Spain. [5]CREAF, Cerdanyola del Vallès, Barcelona 08193, Spain. ✉ e-mail: yejsh@lzu.edu.cn

number of added nutrients increase[9]. Currently, a key knowledge gap is whether and how organic fertilization would potentially increase plant biomass without plant diversity loss. Addressing this question would require a better understanding of the underlying mechanisms of plant diversity loss following fertilization and the differences between organic and inorganic fertilizers in governing these mechanisms.

Three mechanisms have been proposed to explain the plant diversity loss in response to nutrient addition. The biomass-driven competition hypothesis postulates that greater aboveground biomass following inorganic fertilization increases interspecific competition for light, which excludes short-statured plant species[10–12]. The niche dimension hypothesis assumes that a larger number of limiting factors create more niche partitioning opportunities that facilitate species coexistence, whereas fertilization reduces the number of belowground niches[9,11,13]. Finally, the nitrogen detriment hypothesis proposes that long-term nitrogen fertilization may increase the susceptibility of plant species to stress agents[14,15] such as soil acidification[16], ammonium toxicity[17], and soil microbiome change[14]. The relative importance of these mechanisms has yet to be assessed across different environmental conditions such as vegetation properties, soil moisture, and nutrient availability. Indeed, soil water and nutrients influence plant growth and competition, and thus, both could regulate how plant diversity responds to fertilization[18].

Few studies have assessed whether or not organic fertilization also leads to a potential tradeoff between aboveground biomass and plant diversity. Organic fertilizers derived from animal/plant residues and excrement supply organic matter and nutrients to soil, thus increasing soil organic carbon (SOC) storage, cation exchange, water-holding capacity, and microflora[19,20] while decreasing soil evaporative water loss. In contrast, inorganic fertilizers are chemically synthesized and may lack the capacity to improve certain soil properties, such as preventing soil acidification, as effectively as organic fertilizer can[16,21]. Thus, organic fertilization differs from inorganic fertilization in its influence on soil and vegetation conditions. These differences may result in divergent impacts on plant diversity between organic and inorganic fertilizer applications.

The light competition and niche dimension hypotheses both propose that increased aboveground biomass[22] and soil resource availability under organic fertilization should also reduce plant diversity. However, if nitrogen-driven soil acidification and ammonium toxicity were the main mechanisms causing plant diversity loss after nutrient addition, organic fertilization should not result in a decline in plant diversity. As such, organic fertilizers added to grasslands increase biomass and carbon storage without a tradeoff in plant diversity loss. To date, studies have observed that organic fertilization may decrease[16], increase[23], or have no significant effect on plant diversity[22]. However, the number of species in a community is also limited by soil pH[24], moisture[18] and soil fertility[25]. Therefore, the plant diversity response to organic fertilization may depend on local conditions. Assessing the effect of organic fertilization on plant diversity and understanding the underlying mechanisms are thus vital to evaluate tradeoffs between biomass production, soil carbon storage, and plant diversity loss.

Another central role for nature-based solutions is to identify optimal management practices to increase carbon sequestration. A better understanding of carbon sequestration potential following fertilization would need a full assessment of aboveground and belowground responses. SOC, the largest carbon pool in terrestrial ecosystems[26,27], plays a key role in maintaining soil fertility and improving agricultural productivity and sustainability[28]. However, the response of SOC under organic versus inorganic fertilization remains uncertain[29,30] because it may depend on local climate[31], land-use types[32] and nutrient availability[33].

To evaluate the potential for organic fertilizers to be used as a nature-based solution, a full assessment of its benefit is needed across land-use types that often receive additional nutrients to increase production, such as grasslands and croplands. Indeed, organic fertilization has been shown to increase SOC stocks by about 30% in both croplands and grasslands[34,35]. However, due to carbon storage differences between land-use types, the nutrient requirements for carbon sequestration may differ between grasslands and croplands[36]. Soil carbon storage in croplands is also affected by agricultural managements practices[37,38] such as fertilizer rate and tillage, which may lead to different SOC responses to organic fertilization than in grasslands. It remains unclear the degree to which global grasslands and croplands differ in carbon sequestration potential under comparable organic fertilization rates.

To assess the effects of organic versus inorganic fertilization on plant biomass, plant diversity, and SOC, we compiled >5000 pairs (ambient versus fertilization) of measurements from 537 experiments across grasslands and croplands worldwide (Supplementary Fig. 2). We focused on grasslands because a large number of fertilization experiments were conducted in this ecosystem[10]. Moreover, grasslands account for about 41% of the global land surface excluding Greenland and Antarctica. They store around 34% of the terrestrial carbon stock and provide a variety of ecosystem services including forage production, plant diversity conservation, and climate regulation[39]. Croplands are widespread globally, and most agricultural practices reduce soil carbon content[38,40], thus organic fertilizers have the potential to increase SOC under agricultural management practices. We hypothesized that: (1) organic fertilization would increase aboveground biomass more than inorganic fertilization in grasslands, (2) if increased biomass production intensified competition for light, or fertilization reduced belowground niche partitioning, organic fertilization would also cause a decline in plant diversity in grasslands, and (3) if nitrogen detriment (e.g., acidification) was the main mechanism, organic fertilization would not cause plant diversity loss in grasslands. We also evaluated the effect of fertilization on biomass, plant diversity, and SOC across environmental gradients to determine under which conditions tradeoffs among the three ecosystem services would be minimized. Finally, we hypothesized that (4) organic fertilizer added to croplands would lead to comparable increases in SOC relative to grasslands.

## Results
### Global mean responses to fertilization
Inorganic fertilization significantly increased aboveground biomass by 42% ($p < 0.001$) relative to ambient conditions, accompanied by an 18% ($p < 0.001$) decline in species richness and 6% ($p < 0.001$) decline in evenness in semi-natural and natural grasslands (Fig. 1a–c). In contrast, organic fertilization significantly increased aboveground biomass by 56% ($p < 0.001$), without reducing plant diversity relative to ambient conditions (Fig. 1a–c). Compared to inorganic fertilization, organic fertilization significantly increased both species richness and evenness by 10% ($p < 0.001$), while maintaining greater aboveground biomass (Fig. 1a–c). Organic fertilization also significantly increased SOC by 19% ($p < 0.001$) and 15% ($p < 0.001$) compared to ambient conditions and inorganic fertilization, respectively. Inorganic fertilization significantly increased SOC by 2% (Fig. 1d, $p < 0.001$). Thus, although both fertilizers increased aboveground biomass, organic fertilization increased SOC more than did inorganic fertilization while also increasing species richness.

### Fertilization responses across environmental gradients
Based on a multi-model inference procedure for selecting the set of best-fitting models (Supplementary Tables 4–8), we found that the effects of organic fertilization on aboveground biomass depended on climate and soil properties (Fig. 2a). Grasslands with greater mean annual temperature (MAT), soil total nitrogen and fertilizer rate exhibited larger increases in aboveground biomass under organic

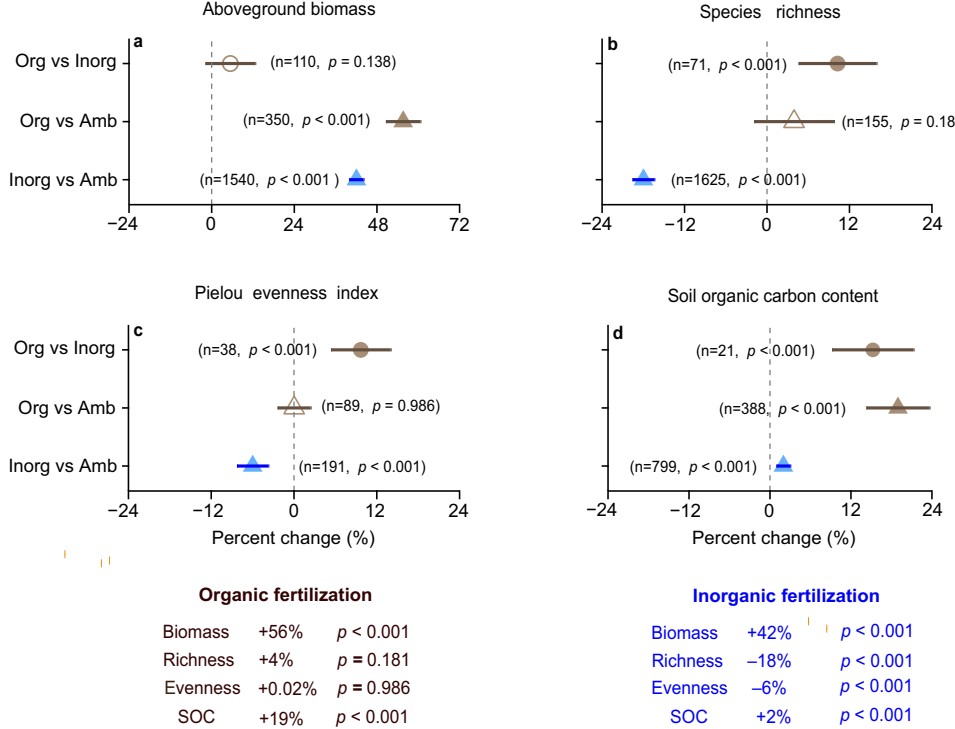

**Fig. 1 | Effects of inorganic and organic fertilization on grasslands plant community and soil organic carbon.** Percent changes in aboveground biomass (**a**), species richness (**b**), Pielou evenness index (**c**) and soil organic carbon (SOC) content (**d**) under organic and inorganic fertilization. Brown and blue triangles represent percent changes under organic and inorganic fertilization relative to ambient conditions, respectively. Brown circles represent percent changes under organic fertilization relative to inorganic fertilization in the same experiments. Filled symbols represent statistical significance when the 95% confidence intervals do not overlap with zero (two-tailed test, $p < 0.05$). The numbers in parentheses show the sample sizes. The sample sizes of SOC are the studies that measured both SOC and aboveground biomass or species richness (**d**). See Supplementary Table 2 for the value of percent changes and confidence intervals. Key: organic fertilization (Org), inorganic fertilization (Inorg), ambient conditions (Amb), aboveground biomass (Biomass), species richness (Richness) and Pielou evenness index (Evenness). Source data are provided as a Source Data file.

fertilization ($p < 0.01$, Fig. 2a–d, and Supplementary Table 9). Similarly, grasslands with greater soil bulk density (SBD), water content, and cation exchange capacity had larger increases in species richness under organic fertilization ($p < 0.05$, Fig. 2e–h, and Supplementary, Table 9). Conversely, under inorganic fertilization, grasslands with greater nitrogen fertilizer rate, number of nutrients added, and soil water content (SWC) showed larger increases in aboveground biomass ($p < 0.05$, Fig. 3a–d, and Supplementary, Table 10). However, increasing inorganic fertilizer rates in grasslands led to larger declines in species richness ($p < 0.001$, Fig. 3e–g, and Supplementary Table 10).

### Relationship between biomass production and plant diversity
Our partial regression and structural equation model (SEM) showed only a weak negative link between the responses of biomass production and species richness under organic fertilization in grasslands (Fig. 4a and Supplementary Fig. 5a). Both organic and inorganic fertilization increased aboveground biomass but only the latter decreased species richness (Fig. 4b and Supplementary Fig. 5b). The SEM indicated that SWC had a positive link with the response of species richness under organic fertilization (Supplementary Fig. 5a), which was consistent with the result of linear mixed effects models (Fig. 2g).

### SOC responses to organic fertilization in grasslands and croplands
In both grasslands and croplands, the effects of organic fertilization on SOC varied with climate, soil properties, and fertilizer rate (Fig. 5b). Organic fertilization increased SOC more in croplands than in

grasslands (32% versus 14% increases, respectively, Fig. 5a) in areas with mean annual temperature (MAT) <15 °C. While in warmer regions, SOC increased more in grasslands than croplands (62% versus 34% increase, respectively, Fig. 5a). After accounting for soil properties and organic fertilizer rate, the effect of organic fertilization on SOC increased significantly with MAT in grasslands ($p < 0.05$, Fig. 5b, c). These results remained consistent when we used a subset (about 40% of all studies) of the data that measured the quality (C:N ratios) of organic fertilizers (Supplementary Fig. S6). Therefore, under a future, warmer climate, applying organic fertilization to grasslands may result in a larger soil carbon sequestration potential than in croplands.

As expected, the effect of organic fertilization on SOC increased with organic fertilizer rate in both grasslands and croplands ($p < 0.05$, Fig. 5b, d). Both the linear mixed effects model (Fig. 5b) and the standardized major axis tests (Fig. 5d and Supplementary Table 13) showed equal SOC increases in grasslands and croplands under comparable organic fertilization rates. The effect of organic fertilization on SOC declined as initial/background SOC in both grasslands and croplands increased, and this pattern was stronger in croplands than in grasslands ($p < 0.05$, Fig. 5b, e). This result has two implications. First, it is not necessary to start with low initial organic carbon for organic fertilizer to increase SOC (Fig. 5e). Second, under similar initial conditions, organic fertilization has a larger effect on SOC in grasslands than in croplands (Fig. 5b).

## Discussion
Using a comprehensive dataset compiled from grassland fertilization experiments worldwide, we found that organic fertilization significantly increased biomass and SOC relative to ambient conditions,

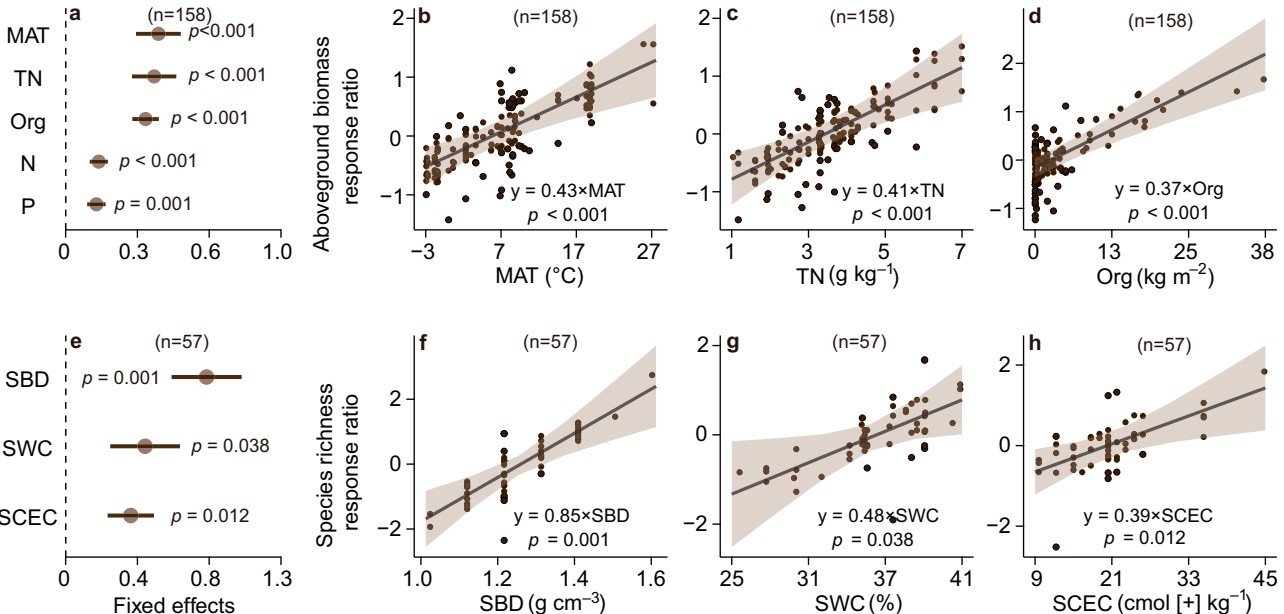

**Fig. 2 | The responses of aboveground biomass and species richness to organic fertilization across environmental gradients. a, e** Brown circles and error bars represent average parameter estimates (standardized regression coefficients) and the 95% confidence intervals in the linear mixed effects models (two-sided), respectively. The numbers in parentheses show the sample sizes. **b–d, f–h** The solid line in each panel shows model fit using partial regression for each environmental factor, and the shading around the fitted line represents the 95% confidence intervals (i.e., error bands represent slopes ± 95% confidence intervals). Equations in **b–d** and **f–h** show the values of standardized regression coefficients. The slopes of the partial regressions are the same as the fixed effects shown in Fig. 2a, e. Complete model statistical results are presented in Supplementary Table 9. Keys: MAT mean annual temperature, Org organic fertilizer amount added, TN soil total nitrogen, N nitrogen fertilizer rate, P phosphorus fertilizer rate, SBD soil bulk density, SCEC soil cation exchange capacity, SWC soil water content. The *p* values are calculated from two-tailed tests. Source data are provided as a Source Data file.

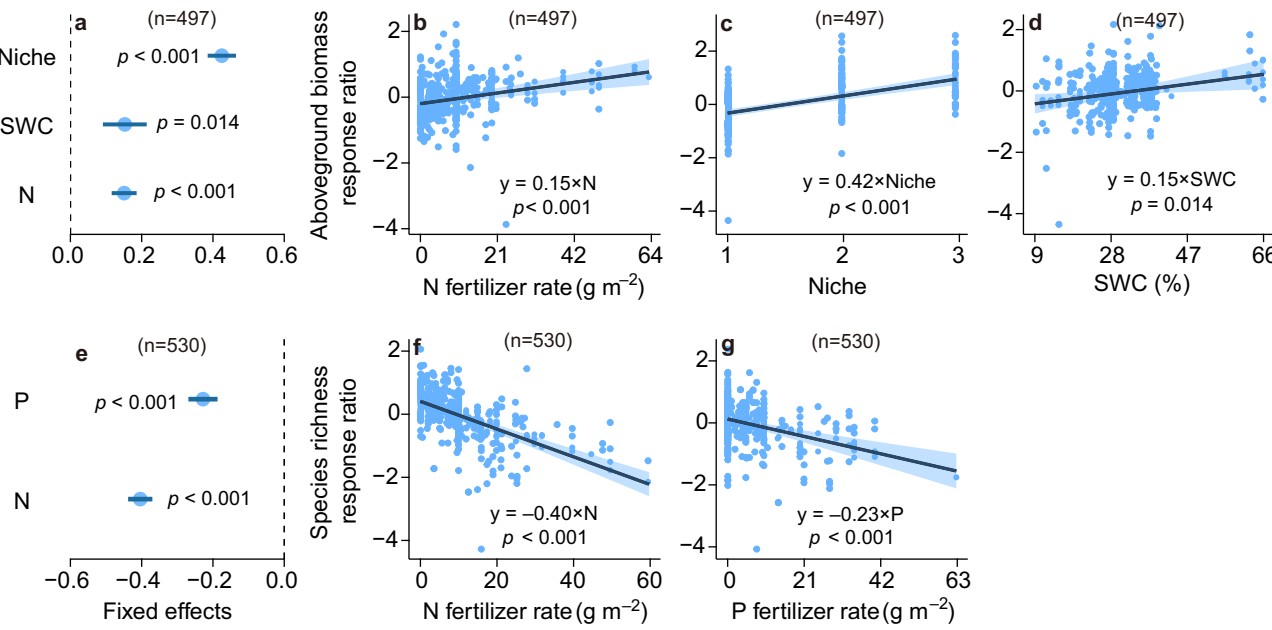

**Fig. 3 | The responses of aboveground biomass and species richness to inorganic fertilization across environmental gradients. a, e** The blue circles and error bars represent average parameter estimates and the 95% confidence intervals in the linear mixed effects models (two–sided), respectively. The numbers in parentheses show the sample sizes. **b–d, f–g** The solid line in each panel shows model fit using partial regression for each environmental factor, and the shading around the fitted line represents the 95% confidence intervals (i.e., error bands represent slopes ± 95% confidence intervals). Equations in **b–d** and **f–g** show the values of standardized regression coefficients. The slopes of the partial regressions are the same as the fixed effects shown in 3a, e. Complete model statistical results are presented in Supplementary Table 10. Keys: Niche number of nutrients added, N nitrogen fertilizer rate, P phosphorus fertilizer rate, SWC soil water content. See Fig. 2 for more details. Source data are provided as a Source Data file.

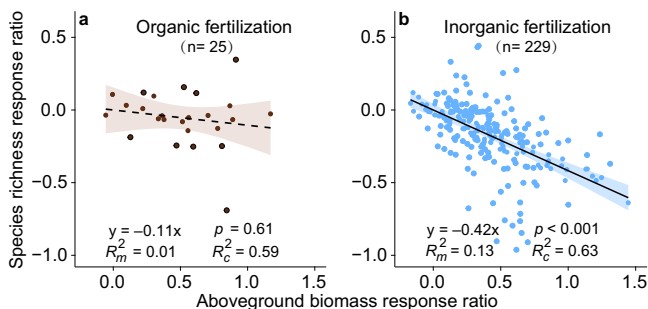

**Fig. 4 | Relationship between the responses of biomass production and species richness under nutrient addition in grasslands.** Field experiments that measured both aboveground biomass and species richness are used in the analyses. **a** Organic fertilization. **b** Inorganic fertilization. We use linear mixed effects models to evaluate the link between the response ratio (ln RR) of biomass and richness (**a**, **b**). The lines in (**a**, **b**) show model fit using partial regression for aboveground biomass, and the shading around the fitted line represents the 95% confidence intervals (i.e., error bands represent slopes ±95% confidence intervals). Solid and dashed lines denote statistically significant (two-sided $p \leq 0.05$) and nonsignificant (two-sided $p > 0.05$) relationships, respectively. The slope of the regression between species richness and aboveground biomass is significantly greater under inorganic fertilization than under organic fertilization ($p_{\text{slope difference}} = 0.023$), according to the Standardized Major Axis Tests. Marginal $R_m^2$ and conditional $R_c^2$ indicate $R^2$ of fixed and random plus fixed effects (**a**, **b**), respectively. The numbers in parentheses show the sample sizes. Complete model statistical results are presented in Supplementary Table 11. Source data are provided as a Source Data file.

with no concomitant plant diversity loss (Fig. 1). Moreover, organic fertilization increased the plant diversity of native species in grasslands with greater soil water content (Fig. 2g). Greater aboveground biomass would increase leaf area, transpiration and canopy interception, and decrease water infiltration[41,42], potentially intensifying plant demand for soil water under fertilization. However, this mechanism may not occur in grasslands with greater soil water content, because water is rarely limiting in ecosystems with greater mean soil moisture. Organic fertilization increased SOC and cation exchange capacity, thus increasing water holding capacity[34,43,44], and decreasing soil water evaporative loss. Therefore, organic fertilization would mitigate biomass-driven declines in soil water, thus avoiding plant diversity loss. Previous studies also supported this finding that mesic grasslands had greater plant species richness than arid grasslands[18,45]. Collectively, these findings support the significant link between soil water and plant diversity in many grasslands.

Our results from either organic or inorganic fertilization in grasslands do not support the niche dimension hypothesis, as the number of nutrients added was not selected in the best fitting models for species richness (Figs. 2e, 3e). The niche dimension hypothesis predicted greater plant diversity loss with an increasing number of added nutrients[9,11,13]. One recent study of inorganic fertilization and water addition tested this hypothesis and found negligible support[10]. Organic fertilizers contain several macro- and micro-nutrients (Supplementary Data 1) but did not reduce plant diversity (Fig. 1), providing no support for niche dimension mechanism. Instead, the components of organic fertilizers would increase soil heterogeneity[46], which may increase the number of micro-niches for plant species, thus reversing the niche dimension hypothesis.

According to the nitrogen detriment hypothesis, inorganic nitrogen increases susceptibility of plants to various stress agents, such as ammonium toxicity, soil acidification, and altered soil microbiome, all of which can lead to plant diversity loss[10,14–17]. We found that both nitrogen and phosphorus fertilizer rates decreased plant diversity (Fig. 3f–g). As such, plant diversity loss under inorganic fertilization might not solely be related to greater nitrogen availability[47]. Indeed, many studies provide robust evidence that inorganic

phosphorus addition leads to species loss[48,49]. We did not find a significant effect of organic fertilization rate on plant diversity (Supplementary Fig. 5a). Organic fertilization improved the soil environment for plant growth by increasing nutrient availability, pH buffering, cation exchange, water holding capacity, and microbial community functioning[43,50,51]. Organic fertilizers need to be decomposed and thus may be less plant-available immediately following application[52,53]. Therefore, organic fertilizer releases nutrients slowly as it decomposes and may continuously provide multiple macro- and micro-nutrients for plant growth[43,54]. According to a subset (about 6% of all studies) of the data that measured both macro- and micro-nutrients (Supplementary Data 1), organic fertilizers supply nitrogen, phosphorus, potassium, magnesium, calcium, zinc, iron, etc. As a result, organic fertilizers increased plant biomass more than did inorganic fertilizers (56% ± 5% versus 42% ± 2%; Fig. 1). A previous study proposed that the average supply rate of the most limiting resource controlled plant diversity[55]. Organic fertilizers provide multiple nutrient resources and thus may avoid plant diversity decline. Collectively, organic fertilization increases soil fertility and aboveground biomass while limiting detrimental effects on plants and microbial diversity.

Organic fertilizers can serve as a nature-based solution to mitigate the tradeoff between increasing aboveground biomass and species loss in grasslands (Fig. 1). Studies showed that the increase in plant diversity under organic fertilization was due to improved soil fertility status, such as water holding capacity and nutrient availability[23,56]. Our finding clearly demonstrates that organic fertilization supports greater native plant diversity compared to inorganic fertilization. Thus, organic fertilization may also facilitate the restoration of degraded grasslands and abandoned croplands by increasing SOC, plant biomass, and plant diversity. However, studies have also reported an increase in exotic plant species with manure application[57]. Thus, we carefully selected and excluded organic fertilization experiments that included exotic plant seeds. As such, the plant diversity response to manure application was not caused by exotic species in our analyses. To avoid introducing exotic plant species and other side-effects, locally-sourced, noncontaminated manure or industrial organic fertilizers should be used as a nature-based solution.

We found that SOC response ratio under organic fertilization increased in grasslands but decreased in croplands under warmer mean annual temperature (Fig. 5c). This difference may result from management practices, where croplands are typically tilled while grasslands are not. Tillage can disrupt soil aggregates and accelerate SOC decomposition by microorganisms[38]. Temperature is also an important factor driving SOC decomposition[58,59]. Tillage in warmer regions may increase SOC decomposition rate, thus leading to a smaller increase in SOC in croplands despite organic fertilization. By contrast, grasslands are rarely tilled. Organic fertilizer increases biomass production in grasslands, which may override the greater loss rate of SOC through decomposition as mean annual temperature increases[60,61]. Irrigation increases soil moisture, and thus affects SOC decomposition and carbon sequestration[62]. Croplands are usually irrigated, while grasslands are not. Warmer temperatures can accelerate soil moisture evaporation, which could slow SOC decomposition in grasslands but not necessarily in croplands. This result suggests that organic fertilization in grasslands may increase soil carbon sequestration potential in a future, warmer climate.

We found that the potential for soil carbon sequestration increased with organic fertilization rate in both grasslands and croplands (Fig. 5b). Considering whole-system feedbacks, increased aboveground productivity following organic fertilization may support more livestock and produce more manure to fertilize grasslands, thus creating a positive feedback loop. However, croplands usually receive imported organic fertilizer from other locations[63], potentially increasing transportation and application costs, which can reduce

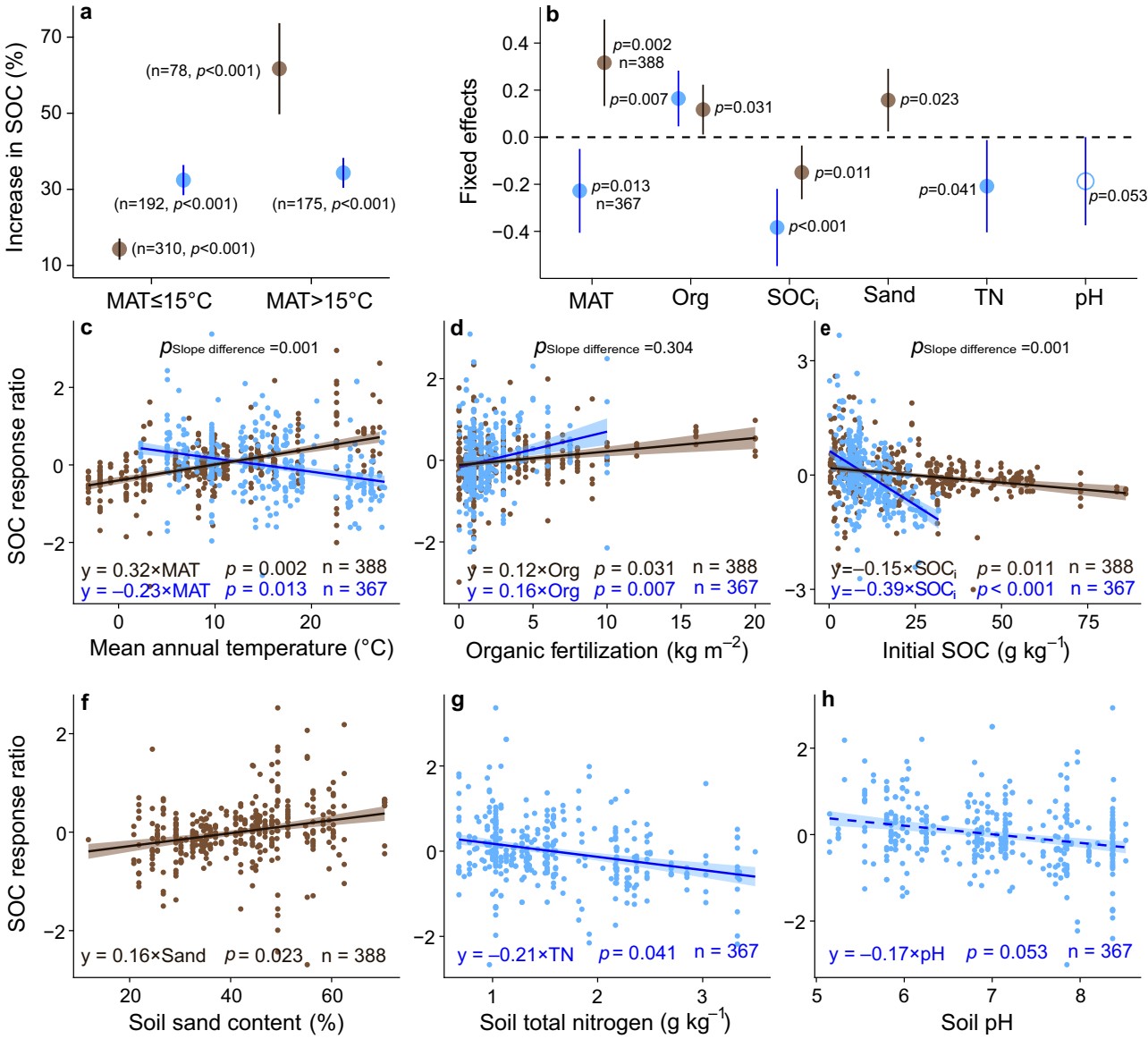

**Fig. 5 | Response of SOC to organic fertilization across environmental gradients in global grasslands and croplands.** Blue and brown represent cropland and grassland, respectively. **a** Percent increase in SOC under organic fertilization in grasslands and croplands. **b** Effects of environmental factors on SOC responses under organic fertilization. **b** Circles and error bars represent average parameter estimates and 95% confidence intervals in the linear mixed effects models (two-sided), respectively. **c**–**h** The brown and blue lines in each panel show model fits using partial regression for each environmental factor, and the shading around the fitted lines represents the 95% confidence intervals (i.e., error bands represent slopes ±95% confidence intervals). The slopes of the partial regressions are the same as the fixed effects shown in **b**. Equations in **c**–**h** show the values of standardized regression coefficients. **a**–**h** The symbol n shows the sample sizes. **c**–**h** Complete model statistical results are presented in Supplementary Tables 6 and 9. Keys: MAT mean annual temperature, Org organic fertilizer amount added, $SOC_i$ initial soil organic carbon, Sand: soil sand content, TN soil total nitrogen, pH soil pH, SBD soil bulk density. The $p$ values are calculated from two-tailed tests. Source data are provided as a Source Data file.

profit margins. We also tested the impact of organic fertilizer quality on SOC response[64,65]. Organic fertilizers with greater carbon:nitrogen ratios led to greater SOC increase in grasslands but less so in croplands (Supplementary Fig. 6), suggesting that low quality organic fertilizers may still promote carbon sequestration in grasslands. Thus, use of organic fertilization in grasslands might be more feasible than in croplands in terms of transportation costs and fertilizer quality.

Collectively, our results highlight the potential of organic fertilization to serve as a nature-based solution that improves grassland ecosystem functions and services, including carbon sequestration, and such potential may actually increase with climate warming in some regions. We found that organic fertilization increased aboveground

biomass production and SOC without a tradeoff in plant diversity loss. By comparison, increased aboveground biomass production and soil carbon storage resulted in plant diversity loss under inorganic fertilization. As such, organic fertilization is favored over inorganic fertilization to improve grassland ecosystem functions and services. Organic fertilization increased plant diversity in grasslands with greater soil moisture. Moreover, organic fertilization may improve the soil environment for plant growth, and thus would avoid plant diversity loss caused by soil acidification and ammonium toxicity that is usually found under inorganic fertilization. Therefore, we argue that increasing the use of organic fertilizers would provide an important nature-based solution to increase productivity and soil carbon sequestration while conserving plant diversity.

## Methods

### Data collection

To compile data on the responses of plant biomass, plant diversity, and SOC to nutrient addition, we searched for peer-reviewed literature published before 30 October 2022 using the web of science and China National Knowledge Network resources. We used the following key-words: (resource addition OR resource availability OR nutrient addition OR nutrient availability OR nitrogen deposition OR nitrogen addition OR nitrogen enrichment OR phosphorus addition OR phosphorus enrichment OR potassium addition OR potassium enrichment OR organic fertilizer OR organic* OR manure* OR farmyard manure* OR pig manure OR cow manure OR horse manure OR sheep manure OR chicken manure OR wet compost) AND (species richness OR plant diversity OR biomass OR aboveground biomass OR AGB OR dry matter yield OR SOC OR soil organic carbon OR SOM OR soil organic matter OR SOC storage) AND (grassland OR meadow OR steppe OR prairie OR herbaceous OR annual OR cropland).

We used three criteria to select literature: (1) field experiments were conducted in semi-natural or natural grasslands, or croplands, and included both ambient and nutrient addition treatments; (2) the means, standard errors or standard deviations and sample sizes were reported; and (3) grassland studies reporting exotic plant species introduced by organic fertilization were excluded (Supplementary Fig. 1). Data were collected from tables in the main text or supporting information when available, or digitally extracted from figures using GetData Graph Digitizer software version 2.26 http://getdata-graph-digitizer.com/.

Our search yielded 537 publications across all continents except Antarctica (Supplementary Fig. 2). In summary, we compiled 1540 pairs (ambient versus fertilization) of field measurements of aboveground biomass, 1625 pairs of species richness, 191 pairs of Pielou evenness index, and 799 pairs of SOC under inorganic fertilization. Under organic fertilization, we compiled 350 pairs of aboveground biomass, 155 pairs of species richness, 89 pairs of Pielou evenness index, 388 pairs of SOC in grasslands and 367 pairs of cropland SOC. Most field experiments in grasslands measured plant aboveground biomass and plant diversity during the peak growing season. In highly managed agroecosystems, only one or two crop species were maintained and thus it is not reasonable to analyze plant diversity. Therefore, we did not collect plant diversity data from cropland.

In our compiled dataset, inorganic/chemical fertilizer included urea (chemically synthesized from inorganic matters), ammonium nitrate, calcium nitrate, super-phosphate, ammonium phosphate dibasic, sodium dihydrogen phosphate, potassium sulphate and potassium chloride. None of the inorganic studies added liquid ammonia fertilizer. Organic fertilization included industrial organic fertilizer, livestock manure and compost. In 12% of field experiments, industrial organic fertilizers were applied, where organic matter was fermented and decomposed at high temperature, and thus any plant seeds within them were killed. The other 88% of experiments applied livestock manure or compost that was usually composted under high temperature and anaerobic environments, and then air-dried to kill plant seeds that may be contained in them. No exotic plant species were reported in all of the organic fertilization field experiments included in our meta-analysis. In 12% of the organic fertilization experiments, inorganic nitrogen, phosphorus, and potassium were also added. About 97% of the field experiments had a duration of <10 years in grasslands, while 74% of the experiments had a duration of >10 years in croplands.

### Meta-analysis

We used the natural log-transformed response ratio (ln RR) to quantify the effect of nutrient fertilization on aboveground biomass, plant diversity, and SOC. The ln RR, also called "effect size", was dimensionless and used to characterize the relative changes between treatment and control[66].

$$\ln RR = \ln\left(\frac{\overline{Y}_t}{\overline{Y}_c}\right) \tag{1}$$

Where $\overline{Y}_t$ and $\overline{Y}_c$ are sample mean values of the response variables (aboveground biomass, or plant diversity, or SOC) in the treatment group (t) and control group (c), respectively. In this study, we calculated two types of response ratio: (1) organic, inorganic fertilization relative to ambient condition (i.e., Org vs. Amb, Inorg vs. Amb) of all experiments, and (2) organic fertilization relative to inorganic fertilization (i.e., Org vs. Inorg) when conducted in the same experiments.

We used a hierarchical model with inverse variance weighting to summarize the response ratio (ln RR) from all individual studies, as this model was usually appropriate for biological experiments[66,67]. Because multiple treatments may share a single control, we added "site" as a random factor in the meta-analysis model to account for non-independence of observations collected from the same site[68,69].

We used the "rma.mv" function in R "metafor" package version 4.4.0 to calculate the weighted mean response ratio (ln RR + +) and the 95% confidence intervals[70]. The 95% confidence intervals were generated by bootstrapping. When they did not overlap with zero, the treatment effects were considered statistically significant.

We applied the Egger's test to examine publication bias[71], and used the trim and fill approach to evaluate the impact of publicaton bias on the meta-anaysis results (Supplementary Table 1 and Fig. 3).

To ease interpretation, we transformed the weighted mean response ratio (ln RR + +) to percent change in the response variables by Eq. 2:

$$\text{Percent change}\,(\%) = \left(e^{\ln RR + +} - 1\right) \times 100\% \tag{2}$$

### Response of biomass, plant diversity and SOC across environment gradients

SOC response ratios related to potential moderators, including mean annual temperature (MAT), soil cation exchange capacity (SCEC), soil total nitrogen (TN), soil pH (pH), soil bulk density (SBD), initial soil organic carbon density, soil sand content (Sand) and soil water content (SWC) and fertilizer rates. We used species richness in this analysis because it had larger sample sizes than other plant diversity indices such as Pielou's evenness (e.g., 155 versus 89 pairs in grasslands). MAT (°C) at each site was extracted from the WorldClim 2 datasets with 30 s spatial resolution[72]. SCEC (cmol [+] kg$^{-1}$), total nitrogen (g kg$^{-1}$), pH, bulk density (g cm$^{-3}$), organic carbon density (kg m$^{-3}$) and sand content (%) at 0–30 cm were extracted from the Soil Grid datasets with 30 s spatial resolution[73]. We used the ERA5-Land datasets of SWC (%) at 0–28 cm with 0.1° spatial resolution, with the mean values calculated during 1982–2022[74].

We used linear mixed effects models to evaluate the response of biomass, plant diversity, and SOC to nutrient fertilization across environmental gradients, with study site as a random effect. We conducted linear mixed models in "lme4" package version 1.1-35.1 and "lmerTest" packages version 3.1.3. To select the set of environmental factors that significantly influenced the response of biomass, plant diversity and SOC to nutrient addition, we conducted a multi-model inference procedure based on the Akaike Information Criterion, using the dredge function in R "MuMIn" package version 1.47.5. To further strengthen the multi-model inference, we also conducted a random forest model to identify the significant environmental predictors of biomass and plant diversity. We used random forest models in the "randomForest" package version 4.7-1.1 to quantify the importance of each predictor, and then used the "rfPermute" package version 2.5.2 to assess the statistical significance of each predictor. The environmental factors selected by the

best linear mixed effects model were also significant predictors identi-fied by a random forest model (Supplementary Fig. 4).

We also employed structural equation model (SEM) to disen-tangle the direct and indirect effects of environmental factors and nutrient fertilization rate on the response ratio (ln RR) of biomass and species richness. We conducted SEM using the R "*lavaan*" package version 0.6-17, and evaluated goodness of the SEM model according to the criteria reported by ref. 75.

To explore whether and where organic fertilizer added to grass-lands led to greater increases in SOC than in croplands, we compared the slopes of the regression of SOC response ratio and environmental drivers (MAT, the amount of organic fertilization added, initial SOC, soil nutrients, soil texture, and pH) between grasslands and croplands, using the Standardized Major Axis Tests and Routines with ordinary least squares regression technique[76].

## Reporting summary

Further information on research design is available in the Nature Portfolio Reporting Summary linked to this article.

## Data availability

All data used in this study, including raw data and source data underlying figures, has been deposited in Figshare https://doi.org/10.6084/m9.figshare.25493419. Mean annual temperature at each site was extracted from the WorldClim database https://www.worldclim.org/. Soil cation exchange capacity, total nitrogen, pH, bulk density, organic carbon density and sand content were extracted from Soil Grid database https://files.isric.org/soilgrids/latest/data_aggregated/1000m/. Soil water content was obtained from ERA5-Land database https://www.ecmwf.int/en/era5-land. Global map was downloaded from natural earth https://www.naturalearthdata.com/. Source data are provided with this paper.

## Code availability

All R code for reproducing the mainly results are available at Figshare https://doi.org/10.6084/m9.figshare.25493419.

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

## Acknowledgements

This research was supported by the Fundamental Research Funds for the Central Universities (lzujbky-2022-ct01) and "111" Project (BP0719040). S.L.C. was supported by NSF award DEB-1856383. J.P. and J.S. were supported by the Spanish Government grants PID2020115770RB-I, PID2022-140808NB-I00, and TED2021-132627 B–I00 funded by MCIN, AEI/10.13039/ 501100011033 European Union Next Generation EU/PRTR, the Fundación Ramón Areces grant CIVP20A6621, and the Catalan Government grants SGR 2021–1333 and AGAUR2023 CLIMA 00118. We thank Prof. Chao Song for suggestions and comments on the writing.

## Author contributions

J.-S.Ye. and T.-S.Shi. conceived this study. T.-S.Shi. and H.-L.Li. collected data. T.-S.Shi. performed the analysis. J.-S.Ye. and T.-S.Shi. drafted the paper. S.L.Collins., K.Yu., J.Peñuelas. and J.Sardans. wrote and edited the final version of the manuscript. All authors contributed critically to the writing and revising of the manuscript.

## Competing interests

The authors declare no competing interests.
