## [Peer Review File · Nature Communications]

Reviewers' Comments:

Reviewer #1:

Remarks to the Author:

GENERAL COMMENTS:

Under review is a global meta-analysis comparing organic and inorganic fertilizer effects on plant biomass, plant diversity, and soil carbon in grasslands. This is a herculean effort and I applaud the authors for attempting such a manuscript. I appreciate how the findings are based in theory, with alternative hypotheses driving the context of organic versus inorganic fertilizer effects on response variables. I also appreciate that the authors followed PRISMA guidelines for the most part. We could use more meta-analyses that follow PRISMA guidelines.

I do have a few major concerns in General Comments that are possible to address with revisions.

GENERAL COMMENTS:

1. The language could use some polishing. There were many spelling/grammar errors. Also overuse of the word 'higher', when in many instances should be greater or more. See Specific Comments below.

2. Figure aesthetics should be improved, especially for journal such as Nature Communications. There are inconsistent color choices, odd symbol usage, panel labeling [a or (a), for example]. The capital and lower case lettering is confusing. Also many figures are too crowded and have excess information or strange abbreviations. I am not convinced that both SEM and random forest regression models are needed. Since SEM takes apriori assumptions, I think random forest is better multi-variate predictor tool. Put the SEM (Figure 4) in the supplemental. See Specific Comments below.

3. Benefits from organic amendments may be from adding other nutrients, not just carbon, that are not added with synthetic, inorganic fertilizer mixtures. A synthesis table of the elements added from all the organic studies, and using this as factor in meta-regression and random forest and SEM models should be added in the supplemental. For example, it could be that adding K, Ca, Mg, Fe, etc.. in manure is the contributing factor to benefits and not carbon. I realize not all studies will have this information but many will. Readers being able to look at a range in these nutrient application rates, including C, in the supplemental section would be very helpful.

4. Why are croplands used in this meta-analysis? More justification and context is needed for this before I read it on Line 250. Authors are also missing some key citations, synthesis, and context from agricultural studies on manure and soil functioning/SOC. It would be helpful to know if and how much other nutrients are also added in the 'inorganic' studies. See references below....

Rui, Y., Jackson, R. D., Cotrufo, M. F., Sanford, G. R., Spiesman, B. J., Deiss, L., ... & Ruark, M. D. (2022). Persistent soil carbon enhanced in Mollisols by well-managed grasslands but not annual grain or dairy forage cropping systems. *Proceedings of the National Academy of Sciences*, 119(7), e2118931119.

Maillard, É., & Angers, D. A. (2014). Animal manure application and soil organic carbon stocks: A meta-analysis. *Global change biology*, 20(2), 666-679.

Kallenbach, C., & Grandy, A. S. (2011). Controls over soil microbial biomass responses to carbon amendments in agricultural systems: A meta-analysis. *Agriculture, Ecosystems & Environment*, 144(1), 241-252.

SPECIFIC COMMENTS:

Title: 16 words is too long and it may be wise to include this is a meta-analysis (although I know

this is cliché). Try to trim down title for Nature Communications. Here is an example: "Organic fertilizers provide multi-functional benefits in grasslands: A global meta-analysis" [10 words]

L23. Delete 'grasslands'.

L23-24. Replace 'caused biodiversity loss relative to ambient' with 'decreased biodiversity'.

L22-25. I suggest rephrasing this without 'actually' and how much, or the percentage increase (and decrease).

L26. Put some bounds by using average and confidence intervals in the abstract perhaps.

L30. Replace 'larger' with 'greater'.

L35-36. Keywords. Don't need to use keywords that are already in the title or abstract. If you don't include 'meta-analysis' in title or abstract, you'll want to definitely include here. Maybe also 'quantitative systematic review'.

L50. 'Nature-based' solutions is vague term. Consider defining and differentiating. Many inorganic fertilizers are, for example, nature-based. Mined P and K for example, are the major sources of 'inorganic' fertilizers.

L64. Delete 'highly'.

L89. How does it differ? Some explanation is needed.

L97. Add hyphen between 'nitrogen' and 'driven'. Is soil toxicity the right word here? I think the soil just becomes acidic. How many of the studies add liquid ammonia fertilizer that would cause burn?

L112-113. Odd phrasing. Rephrase sentence with 'Organic fertilizers can...'. This idea of the C inputs should be considered as part of the reason there are increases in SOC. A rate of C applied should be determined for each study and used as factor in analyses. Also, the other nutrients being added [See General Comment #3].

L120-121. It is unclear what the authors mean by 'carbon distribution patterns'. Please re-write this sentence for clarity.

L123. Replace 'response of SOC' with 'SOC responses'.

L124. Delete 'between land-use types'.

L164. The triangles and color patterns do not look 'yellow'. They look orange by my eyes. I would suggest the authors use different color scheme among other improvements in figures [See General Comment #2]. I suggest using colors that are intuitive. Use light brown or dark brown for organic fertilizer. Use blue or other color that can be differentiated from brown by color-blind folks for the inorganic fertilizer. See the many color-blind friendly palettes available in R.

L178. Replace 'amount of nutrient added' with 'fertilizer rate'. Use different abbreviations than what is used in the figures - "N and P". It is not specific enough. Use "N fertilizer rate" or "P fertilizer rate" in Figure 3d and 3e (Line 201-202).

L185. Avoid the use 'higher' to describe an amount. Higher tends to imply positionality of something. In other words some thing being above another thing. Use 'greater' if you're talking about a quantity or amount.

L230. I don't understand the colors in Figure 4B and 4D. Why are these orange, the same color as inorganic fertilization? Then there are blue colors that also correspond to organic in both panels. See General Comment #2 about improving figure aesthetics.

L250. More context is needed here – I thought this was a paper about grasslands and fertilizer additions. Why is there all the sudden cropland included in comparison? This was not mentioned prior to this point. I think I understand the authors reason for comparing but it needs to be made clear at the beginning of the paper.

L273. Too many significant digits in coefficients. Also need multiplication sign on slopes.

L294. Avoid 'higher' when talking about a quantity.

L315. Delete 'empirical'. Perhaps the authors mean 'Individual'. I would hope all of the studies are empirical.

L331. Replace 'amounts of nitrogen and phosphorus added' with 'greater fertilizer rate'

L339-340. Re-write this sentence. Organic fertilizers are not 'slow acting' but do contain other macro and micronutrients. Some of the nutrients need to be mineralized and thus may be less plant-available immediately.

L346. Replace 'Empirical'

L350. Avoid 'higher'

L370. 'lateral transfer of organic fertilizer' seems like jargon. Do you mean 'imported' from a different location?

L414. I like the flow diagram for paper selection process. But the authors should put in the diagram what criteria were used at which stage.

Reviewer #2:

Remarks to the Author:

Shi et al. conducted a meta-analysis of results from 424 field experiments of organic and inorganic fertilization on plant productivity (biomass) and diversity, and soil organic C. They found that both fertilization practices significantly increased plant biomass, as expected, but have different effects on plant diversity and soil organic C. Organic fertilization increased soil organic C and did not induce a tradeoff between biomass and diversity, which is the most important finding. In contrast, inorganic fertilization reduced plant diversity and had no positive effect on soil organic C. In particular, in the warmer region, organic fertilization increased soil organic C in grasslands but not in croplands. Based on these results, they proposed that water preservation under organic fertilization may play a major role in preserving plant diversity in grasslands, and applications of organic fertilizers can increase plant biomass and soil organic C, without inducing biomass-diversity tradeoffs.

These results are interesting, and the manuscript is well written. The analyses of data applied basically follow the routine methods for meta-analyses. However, there are several issues that need to be clarified or addressed before this manuscript can be accepted for publication.

Major comments:

FIRST, the comparison of the impact of organic fertilization on soil organic C between grasslands and croplands could be misleading: both the quantity and quality of the organic inputs can significantly affect the C retention, and stricter requirements for both the quantity and quality are often applied for croplands than grasslands. Were higher soil C benefits in grasslands due to higher amount but lower quality of organic input than in croplands?

Quantity of organic inputs: High organic input can negatively affect annual crop plants (seedlings in particular), but may have limited effects on perennial grasses. Also, it is well-known that low

levels of organic C input to croplands may increase crop yields or plant biomass but only have marginal impact on soil organic C. Fig. 5b showed that some organic inputs to grasslands (over 20 g m²) were much higher than those to croplands (the maximum was at 10 g m²). Simply looking at the two regression lines, however, figure 5b seems to suggest that SOC in croplands may be more responsive to high organic fertilization. Then, if similar amounts of organic input were applied, would the C benefit still differ between grasslands and croplands?

In Fig. 5c, effect of initial soil C: the difference in the slope of two regression lines (one for croplands and another for grasslands) can be largely attributed to one data point of a high SOC in croplands: that data point has a SOC of ca. 10%, which is very different from most global croplands. Is that fair to draw a general conclusion? I understand it is very challenging to deal with this kind of situation, but it may need to mention this in the Discussion.

Quality of organic inputs: Low quality (e.g., high C:N ratio with low contents of available nutrients) of organic fertilizers can have negative effects on crop plants (saying wheat plants) but may still promote growth of some grasses (e.g., switchgrass) because some grasses have lower demands for nutrients. Yet, quality of organic input affects its decomposition and retention in soil.

SECOND, the diversity issue: Yes, organic fertilization may maintain or even enhance plant diversity, but this may have different implications for croplands, intensively managed forage fields, and semi-natural grasslands. For a field of crop plants or forage plants, high diversity likely means more weeds, which may compromise both the yield and quality of a crop or forage plants. In contrast, high diversity may be beneficial in natural grasslands. In other words, high diversity may be good for conservation (and other ecosystem services) but does not necessarily mean it is desired in highly managed agroecosystems. May the data further be analyzed to examine the differences among croplands, intensively managed forage fields, and semi-natural grasslands?

THIRD, general benefits of organic vs. inorganic fertilization: One major message of the manuscript is that organic fertilization may have more benefits in terms of both plant diversity/biomass and soil organic C and provides a climate-friendly option for grassland management. The benefit of organic fertilization is not controversial, and I believe, is also accepted by many farmers and land managers. However, the challenge is that organic fertilization has some problems or disadvantages too. One of them is that the economic cost of transporting and spreading organic fertilizers is much higher than that for inorganic fertilizers, which cuts into profit margins of farming as an industry. This issue may need to be discussed while recommending organic fertilization.

Minor edits or typos:

Line 83: mechanisms.

Figures: the labeling of several figures is confusing. For example, Figure 2 has A and B, as well as a, b, c, d, e, and f. It seems there were some different connections between A (or B) and a, b, c, d, e, and f. May Fig. 2A, Fig.2A1, Fig.2A2 and Fig.2A3 work? Similar situation for Figure 3.

Lines 275-276: Effects of

Lines 370-371: not a complete sentence.

Response to Reviewers

Reviewer 1

Under review is a global meta-analysis comparing organic and inorganic fertilizer effects on plant biomass, plant diversity, and soil carbon in grasslands. This is a herculean effort and I applaud the authors for attempting such a manuscript. I appreciate how the findings are based in theory, with alternative hypotheses driving the context of organic versus inorganic fertilizer effects on response variables. I also appreciate that the authors followed PRISMA guidelines for the most part. We could use more meta-analyses that follow PRISMA guidelines.

I do have a few major concerns in General Comments that are possible to address with revisions.

Responses: We are thankful for the comprehensive and encouraging evaluation, which greatly improved our manuscript. We have carefully revised the manuscript accordingly. Please see our point-by-point responses below.

General comments:

Comment #1. The language could use some polishing. There were many spelling/grammar errors. Also overuse of the word ‘higher’, when in many instances should be greater or more. See Specific Comments below.

Responses: In the revision, we polished the language and corrected the spelling/grammar errors. We also changed ‘higher’ to ‘greater’ or ‘more’ when necessary and appropriate. For example, in Line 195 and in Line 298.

Comment #2. Figure aesthetics should be improved, especially for journal such as Nature Communications. There are inconsistent color choices, odd symbol usage, panel labeling [a or (a), for example]. The capital and lower case lettering is confusing. Also many figures are too crowded and have excess information or strange abbreviations. I am not convinced that both SEM and random forest regression models are needed. Since SEM takes apriori assumptions, I think random forest is better multi-variate predictor tool. Put the SEM (Figure 4) in the supplemental. See Specific Comments below.

Responses: Thank you for the suggestion for improving the figures. In the revision, we have revised and used consistent color, symbol and panel labeling for all figures.

Following your suggestion, we have now moved the SEM figure to SI Appendix as Figure S5 in the revision.

Comment #3. Benefits from organic amendments may be from adding other nutrients, not just carbon, that are not added with synthetic, inorganic fertilizer mixtures. A synthesis table of the elements added from all the organic studies, and using this as a factor in meta-regression and random forest and SEM models should be added in the supplemental. For example, it could be that adding K, Ca, Mg, Fe, etc., in manure is the contributing factor to benefits and not carbon. I realize not all studies will have this information but many will. Readers being able to look at a range in these nutrient application rates, including C, in the supplemental section would be very helpful.

Responses: Thanks for this constructive and insightful suggestion.

We have now carefully searched and compiled data of other nutrients (i.e., K, Ca and Mg, etc.) from the organic fertilization studies that provided this information. We have found that less than 6% of organic fertilization studies in grasslands and croplands measured micronutrients. Thus, the micronutrients data were not sufficient for including in the linear mixed model, random forest model and SEM.

In the revision, we have, however, added a summary table to show that the micronutrients were added in the organic fertilization experiments (Table R1). We also included the ranges of these micronutrients in the Discussion (Lines 347-348).

Table R1 Summary table of micronutrients measured in organic fertilizers in this meta-analysis. The different values are the amount resulting from distinct doses of organic inputs.

Id	Title	Type	Micronutrients contained in organic fertilizer
1	Effect of cattle slurry on soil and herbage chemical properties, yield, nutrient balance and plant species composition of moderately dry Arrhenatherion grassland	Cattle slurry	Ca(g/m ²):2.2, 4.4, 6.6, 8.8, Mg(g/m ²):0.9, 1.8, 2.7, 3.6
2	Effects of organic fertilizer on soil nutrient status, enzyme activity, and bacterial community diversity in Leymus chinensis steppe in Inner Mongolia, China	Compost	Available P:332.1 mg/kg, available K:1765 mg/kg
3	Effects of organic fertilizer return on soil nutrients in temperate meadow steppe (in Chinese)	Compost	Available N:790 mg/kg, available P:110 mg/kg, available K:3250 mg/kg
4	The effect of root and fertilization on vegetation characteristics and nutrient content of Leymus chinensis meadow (in Chinese)	Farmyard manure	N:6%, P:6%, K:6%
5	Microbial biomass, organic matter mineralization and nitrogen in soils from long-term experimental grassland plots (Palace Leas meadow hay plots, UK)	Farmyard manure	C:40g/m ² , S:1.1 g/m ²
6	Effects of 5 years of digestate application on biomass production and quality of cocksfoot (Dactylis glomerata L.)	Digestate manure	SOC:44.34 g/kg, Cd:1.182mg/kg, Cr:13.62 mg/kg, Ni:16.96, Pb:9.74 mg/kg, Cu:747.6 mg/kg, Zn:929.4 mg/kg, ph:8.08
7	Effects of fertilizations on soil bacteria and fungi communities in a degraded arid steppe revealed by high through-put sequencing	Sheep manure, CO(NH ₂) ₂ , Ca(H ₂ PO ₄) ₂ H ₂ O	Available P:0.0812mg/kg Available K:0.0164 mg/kg
8	The Effect of Fertilizers on Biomass and Biodiversity on a Semi-Arid Grassland of Northern China		N:4.2 g/m ² , P:2.1 g/m ² , SOM:63g/m ² , N:8.4 g/m ² , P:4.2 g/m ² , SOM:126 g/m ² , N:12.6 g/m ² , P:6.3 g/m ² , SOM:189 g/m ²
9	Effects of organic fertilizer on plant species diversity and biomass of common species of Leymus chinensis steppe (in Chinese)	Farmyard manure	SOM:24.61%, 26.085%, Availablep:382.01mg/kg,282.23mg/kg Availablek:690.7mg/kg,2839.2 mg/kg Moisture:1.01%, 0.98%
10	Root distribution and herbage production under different management regimes of mountain grassland	Organic fertilizer	6.9 g N kg ⁻¹ , 1.4 g P kg ⁻¹ , 6g K kg ⁻¹ ,
11	Sustainable Management of Nardus stricta L. Grasslands in Romania's Carpathians	Farmyard manure	0.42% N, 0.19%P, 0.27%K
12	Pig slurry in carpet grass pasture: Yield and plant-available nitrogen	Farmyard manure	(kg m ⁻²): P:0.8, K:0.6, Ca: 0.7, Mg: 0.4, Zn:37.9, Mn:16.8, pH:7.4
13	Long-term alternative dairy manure management approaches enhance microbial biomass and activity in perennial forage grass	Dairy manure, NH ₄ NO ₃	Available K: 12.6 g/kg, 14 g/kg, 16.4 g/kg, 17 g/kg,33.1 g/kg, 37.5 g/kg,
14	Different effects of sheep excrement type and supply level on plant and soil C:N:P stoichiometry in a typical steppe on the loess plateau	Sheep manure	0.041 g N m ⁻² 0.026 g P m ⁻² 0.081 g N m ⁻² 0.051 g P m ⁻² 0.123 g N m ⁻² 0.077 g P m ⁻²
15	Comparative effects of food processing liquid slurry and inorganic fertilizers on tanner grass (Brachiaria arrecta) pasture: grass yield, crude protein and P levels and residual soil N and P	Liquid slurry	0.05 g N m ⁻² 0.2 g N m ⁻² 0.0381 g P m ⁻² 0.0531 g P m ⁻²
16	Effects of cow dung and wood biochars and green manure on soil fertility and tiger nut (Cyperus esculentus L.) Performance on a savanna Alfisol	Cow dung, wood biochar, green manure	Ca:13.8g/m ² , Mg:13.6g/m ² , Na:18.8 g/m ² , K:36.6 g/m ² , Ca:12.4g/m ² , Mg:4.1g/m ² , Na:6.8 g/m ² , K:14.1

			g/m ² , Ca:34.2g/m ² , Mg:1.1g/m ² , Na:1.4 g/m ² , K:44.1 g/m ²
17	Effect of sewage sludge on metal content of grassland soil and herbage in semiarid lands	Sewage sludge	SOM:25%, pH:6.2, Fe:12.71mg/kg, Mn:74.2 mg/kg, Mg:270 mg/kg, Cu: 11.16mg/kg, Pb:0.98 mg/kg, Ca:0.44 mg/kg,
18	The soil microbial community in a sewage-sludge-amended semi-arid grassland	Sewage sludge	Na:587mg/kg, Ca:58 mg/kg, Mg:21mg/kg, K:770 mg/kg, P:1599 mg/kg, Zn:150 mg/kg, Fe:259.2 mg/kg, Mn:11.8 mg/kg, Cu:47.5 mg/kg, Cd:0.24 mg/kg, Pb:3.3 mg/kg, Al:0.1ug/g, B:0.1 mg/kg
19	Composition of fungal groups associated with sewage sludge amended grassland soils	Sewage sludge	Na:587mg/kg, Ca:58 mg/kg, Mg:21mg/kg, K:770 mg/kg, P:1599 mg/kg, Zn:150 mg/kg, Fe:259.2 mg/kg, Mn:11.8 mg/kg, Cu:47.5 mg/kg, Cd:0.24 mg/kg, Pb:3.3 mg/kg, Al:0.1ug/g, B:0.1 mg/kg
20	Soil restoration in semiarid Patagonia: Chemical and biological response to different compost quality	Compost	SOC:211g/kg, 128g/kg, N: 15.5g/kg, 7.7g/kg, Ca: 12.4g/kg, 13.7g/kg, Mg: 2.7g/kg, 6.1g/kg, K: 1.7g/kg, 1.5g/kg, Na: 0.7g/kg, 0.9g/kg P: 5.2g/kg, 4g/kg
21	Biowaste effects on soil and native plants in a semiarid ecosystem	Compost	SOC:332g/kg, P:3.7 g/kg, K:5 g/kg, Cd:1.48 mg/kg, Cu:203 mg/kg, Cr:32.9 mg/kg, Pb:193 mg/kg, Ni:21.6 mg/kg, Zn:334 mg/kg
22	Soil Organic Carbon Sequestration by Long-Term Application of Manures Prepared from Trianthema portulacastrum Linn	Farmyard manure, Compost	Ca:1%
23	Ten-year long-term organic fertilization enhances carbon sequestration and calcium-mediated stabilization of aggregate-associated organic carbon in a reclaimed Cambisol	Farmyard manure	25.8%SOC, 1.68%N, 1.54%P ₂ O ₅ , 0.82%K ₂ O, 4.98%Ca, pH:7.9

Comment #4. Why are croplands used in this meta-analysis? More justification and context is needed for this before I read it on Line 250. Authors are also missing some key citations, synthesis, and context from agricultural studies on manure and soil functioning/SOC. It would be helpful to know if and how much other nutrients are also added in the ‘inorganic’ studies. See references below....

Rui, Y., Jackson, R. D., Cotrufo, M. F., Sanford, G. R., Spiesman, B. J., Deiss, L., ... & Ruark, M. D. (2022). Persistent soil carbon enhanced in Mollisols by well-managed grasslands but not annual grain or dairy forage cropping systems. *Proceedings of the National Academy of Sciences*, 119(7), e2118931119.

Maillard, É., & Angers, D. A. (2014). Animal manure application and soil organic carbon stocks: A meta - analysis. *Global change biology*, 20(2), 666-679.

Kallenbach, C., & Grandy, A. S. (2011). Controls over soil microbial biomass responses to carbon amendments in agricultural systems: A meta-analysis. *Agriculture, Ecosystems & Environment*, 144(1), 241-252.

Responses: Thank you for the comment and helpful suggestion.

We have now added more justification and context for using croplands in this meta-analysis. A novel question might be to explore whether and where organic fertilizer added to grasslands led to greater increases in soil organic carbon than in croplands. (Lines 127-130, and Lines 150-152). We found that the positive effect of organic fertilization on SOC increased with increasing mean annual temperature in grasslands, a pattern which was not found in croplands. Thus, in a future, warmer climate, organic fertilization in grasslands may have greater soil carbon sequestration potential than when used in croplands.

We have now included these key citations in the revision (Lines 113-122, and 384 in the main text).

We have also added a table providing information on whether and how much other nutrients are also added in the inorganic fertilization experiments. See *SI Appendix*, Table S14.

SPECIFIC COMMENTS:

Comment #5. Title: 16 words is too long and it may be wise to include this is a meta-analysis (although I know this is cliché). Try to trim down title for Nature Communications. Here is an example: “Organic fertilizers provide multi-functional benefits in grasslands: A global meta-analysis” [10 words]

Responses: Thanks for your helpful suggestion. We revised the title accordingly (Line 1)

Comment #6. L23. Delete ‘grasslands’.

Responses: Change made accordingly. See lines 22-23

Comment #7. L23-24. Replace ‘caused biodiversity loss relative to ambient’ with ‘decreased biodiversity’.

Responses: Change made accordingly. See line 24

Comment #8. L22-25. I suggest rephrasing this without ‘actually’ and how much, or the percentage increase (and decrease).

Responses: Change made accordingly. See line 24-25.

Comment #9. L26. Put some bounds by using average and confidence intervals in the abstract perhaps.

Responses: We added some bounds in the abstract. See lines 23 and lines 26-27.

Comment #10. L30. Replace ‘larger’ with ‘greater’.

Responses: Change made accordingly. See line 30.

Comment #11. L35-36. Keywords. Don’t need to use keywords that are already in the title or abstract. If you don’t include ‘meta-analysis’ in title or abstract, you’ll want to definitely include here. Maybe also ‘quantitative systematic review’.

Responses: We have now excluded keywords that are already in the abstract or title. See lines 35-36.

Comment #12. L50. ‘Nature-based’ solutions is vague term. Consider defining and differentiating. Many inorganic fertilizers are, for example, nature-based. Mined P and K for example, are the major sources of ‘inorganic’ fertilizers.

Responses: Many thanks for your comment.

Nature-based solutions are defined by International Union for Conservation of Nature (IUCN) as “actions to protect, sustainably manage, and restore natural or modified ecosystem, that address societal challenges effectively and adaptively, simultaneously providing human well-being and biodiversity benefits”¹.

Organic fertilization can improve soil fertility and increase aboveground biomass while avoiding the negative impacts on soil conditions for most plants and microbes^{2,3}. Inorganic fertilizer can potentially cause species decline and N₂O emissions, therefore, we understand that it is not a nature-based solution. In any case, we have now revised the sentence to read: “Nature based solutions (i.e., such as organic fertilization and

avoiding over-application of inorganic fertilizer) offer the potential to guide actions to improve land management practices and mitigate the consequences of global environmental change, such as enhancing carbon sequestration and reducing biodiversity loss^{4,5}". See lines 39-43.

Comment #13. L64. Delete 'highly'.

Responses: Change made accordingly. See line 55

Comment #14. L89. How does it differ? Some explanation is needed.

Responses: We have now explained the difference in the revision. It now reads: "Organic fertilizers derived from animal, plant residues, as well as excrement, play a crucial role in supplying organic matter and nutrients to soil. They contribute to the enhancement of soil organic carbon, cation exchange and water-holding capacity, and decrease soil water evaporative loss^{2,3,6}. In contrast, inorganic fertilizers, also known as mineral fertilizers, are chemically synthesized or mined and may lack the capacity to improve certain soil properties, such as preventing soil acidification, as effectively as organic fertilizer can⁷⁻⁹." See lines 78-86.

Comment #15. L97. Add hyphen between 'nitrogen' and 'driven'. Is soil toxicity the right word here? I think the soil just becomes acidic. How many of the studies add liquid ammonia fertilizer that would cause burn?

Responses: You are right. We have revised the sentence. It now reads: "Therefore, if nitrogen-driven soil acidification and ammonium toxicity were the main mechanism causing biodiversity loss after nutrient addition, organic fertilization may not result in a decline in biodiversity." See lines 93-95.

In our compiled dataset, inorganic fertilizer included urea, ammonium nitrate, calcium nitrate, super-phosphate, ammonium phosphate dibasic, sodium dihydrogen phosphate, potassium sulphate and potassium chloride. Thus, all inorganic studies did not add liquid ammonia fertilizer. We have clarified this information in the Methods section (Lines 444-446).

Comment #16. L112-113. Odd phrasing. Rephrase sentence with 'Organic fertilizers can...'. This idea of the C inputs should be considered as part of the reason there are increases in SOC. A rate of C applied should be determined for each study and used as factor in analyses. Also, the other nutrients being added [See General Comment #3].

Responses: Thanks for your suggestion. We have revised the sentence. It now reads: "Organic fertilizers release nutrients slowly as they decompose and may continuously

provide a balanced supply of multiple nutrients for plant growth¹⁰⁻¹²”. See lines 109-111.

Comment #17. L120-121. It is unclear what the authors mean by ‘carbon distribution patterns’. Please re-write this sentence for clarity.

Responses: We clarified the sentence. It now reads: “Soil carbon storage in croplands is affected by climate¹³ (i.e., temperature), soil properties¹⁴ (i.e., pH, texture and initial SOC) and agricultural managements practices^{15,16} (i.e., fertilizer rate and tillage method), all of which may led to different SOC responses to organic fertilization than in grasslands”. See lines 124-127.

Comment #18. L123. Replace ‘response of SOC’ with ‘SOC responses’.

Responses: Change made accordingly. See line 127

Comment #19. L124. Delete ‘between land-use types’.

Responses: Change made accordingly. See line 127

Comment #20. L164. The triangles and color patterns do not look ‘yellow’. They look orange by my eyes. I would suggest the authors use different color scheme among other improvements in figures [See General Comment #2]. I suggest using colors that are intuitive. Use light brown or dark brown for organic fertilizer. Use blue or other color that can be differentiated from brown by color-blind folks for the inorganic fertilizer. See the many color-blind friendly palettes available in R.

Responses: Change made accordingly. See Figure R1.

Figure R1. Percent changes in aboveground biomass (a), species richness (b), Pielou evenness index (c) and soil organic carbon content (d) under organic and inorganic fertilization.

Comment #21. L178. Replace ‘amount of nutrient added’ with ‘fertilizer rate’. Use different abbreviations than what is used in the figures – “N and P”. It is not specific enough. Use “N fertilizer rate” or “P fertilizer rate” in Figure 3d and 3e (Line 201-202).

Responses: Change made accordingly. See line 188, line 212 and line 221. We changed “N and P” to “N fertilizer rate” and “P fertilizer rate” in Figure 3b and 3f (Line 215), respectively. See Figure R2.

Figure R2. The responses of aboveground biomass and species richness to inorganic fertilization across environmental gradients.

Comment #22. L185. Avoid the use ‘higher’ to describe an amount. Higher tends to imply positionality of something. In other words, some thing being above another thing. Use ‘greater’ if you’re talking about a quantity or amount.

Responses: Change made accordingly. See line 195

Comment #23. L230. I don’t understand the colors in Figure 4B and 4D. Why are these orange, the same color as inorganic fertilization? Then there are blue colors that also correspond to organic in both panels. See General Comment #2 about improving figure aesthetics.

Responses: Please see our response to Comment #2.

Comment #24. L250. More context is needed here – I thought this was a paper about grasslands and fertilizer additions. Why is there all the sudden cropland included in comparison? This was not mentioned prior to this point. I think I understand the authors reason for comparing but it needs to be made clear at the beginning of the paper.

Responses: Thank you for the valuable suggestion. Please see our response to Comment #4.

Comment #25. L273. Too many significant digits in coefficients. Also need multiplication sign on slopes.

Responses: Thank you. We kept two significant digits and added multiplication sign

on slopes, in the revision. For example, in Line 198, in Line 215.

Figure R3. The responses of aboveground biomass and species richness to organic fertilization across environmental gradients.

Figure R4. The responses of aboveground biomass and species richness to inorganic fertilization across environmental gradients.

Comment #26. L294. Avoid ‘higher’ when talking about a quantity

Responses: We changed it to ‘greater in the revision. See line 298.

Comment #27. L315. Delete ‘empirical’. Perhaps the authors mean ‘Individual’. I would hope all of the studies are empirical.

Responses: Change made accordingly. See line 319.

Comment #28. L331. Replace ‘amounts of nitrogen and phosphorus added’ with ‘greater fertilizer rate’

Responses: Change made accordingly. See line 335.

Comment #29. L339-340. Re-write this sentence. Organic fertilizers are not ‘slow acting’ but do contain other macro and micronutrients. Some of the nutrients need to be mineralized and thus may be less plant-available immediately.

Responses: Thanks for the comments. We revised these sentences and clarified.

“Organic fertilizers contain macro- and micronutrients, some of which need to be mineralized and thus may be less plant-available immediately^{17,18} (Lines 343-345). Therefore, organic fertilization differs from inorganic fertilization in its influencing soil and vegetation conditions. Their dissimilarity may result in divergent impacts on plant biodiversity and soil carbon storage compared to inorganic fertilization. The hypotheses of light competition and niche dimension propose that the increase in aboveground biomass^{12,19,20} and soil resource availability under organic fertilization might lead to a reduction in plant biodiversity. However, the application of organic fertilizers would enhance soil organic carbon (SOC), water-holding capacity, and microflora^{21,22}. This, in turn, helps counteract the negative effects of soil acidification and ammonium toxicity caused by inorganic nitrogen.

Comment #30. L346. Replace ‘Empirical’

Responses: Thank you. Agreed and done. See line 355.

Comment #31. L350. Avoid ‘higher’

Responses: Thank you. Agreed and done. See line 358.

Comment #32. L370. ‘lateral transfer of organic fertilizer’ seems like jargon. Do you mean ‘imported’ from a different location?

Responses: You are right. We changed it to ‘imported organic fertilizer from other location’. See lines 389-391.

Comment #33. L414. I like the flow diagram for paper selection process. But the authors should put in the diagram what criteria were used at which stage.

Responses: Thank you very much for your suggestion. We included a revised diagram in Figure S1.

Figure R5. Flow diagram showing the process of literature search in this study.

Replies to Reviewer 2

Shi et al. conducted a meta-analysis of results from 424 field experiments of organic and inorganic fertilization on plant productivity (biomass) and diversity, and soil organic C. They found that both fertilization practices significantly increased plant biomass, as expected, but have different effects on plant diversity and soil organic C. Organic fertilization increased soil organic C and did not induce a tradeoff between biomass and diversity, which is the most important finding. In contrast, inorganic fertilization reduced plant diversity and had no positive effect on soil organic C. In particular, in the warmer region, organic fertilization increased soil organic C in grasslands but not in croplands. Based on these results, they proposed that water preservation under organic fertilization may play a major role in preserving plant

diversity in grasslands, and applications of organic fertilizers can increase plant biomass and soil organic C, without inducing biomass-diversity tradeoffs.

These results are interesting, and the manuscript is well written. The analyses of data applied basically follow the routine methods for meta-analyses. However, there are several issues that need to be clarified or addressed before this manuscript can be accepted for publication.

Responses: We are thankful for the comprehensive and encouraging positive evaluation, which greatly improved our manuscript. We have carefully revised the manuscript accordingly, please see our point-by-point responses below.

Major comments:

Comment #34. FIRST, the comparison of the impact of organic fertilization on soil organic C between grasslands and croplands could be misleading: both the quantity and quality of the organic inputs can significantly affect the C retention, and stricter requirements for both the quantity and quality are often applied for croplands than grasslands. Were higher soil C benefits in grasslands due to higher amount but lower quality of organic input than in croplands?

Responses: Thank you for your insightful comment. We agree that it is vital to consider the impacts of the quantity and quality of organic input on SOC.

First, please note that we had carefully considered the impacts of the amounts of nutrient added on SOC through using linear mixed effect models and partial regression analysis. Our partial regression analysis suggested that there was no significant difference in the increasing rate of SOC when similar amount of organic fertilizer was applied to grasslands and croplands ($p=0.304$, Figure R6d). After accounting for soil properties and the organic fertilization amount, the effect of organic fertilization on SOC increased with increasing mean annual temperature in grasslands, but decreased as temperature increased in croplands ($p=0.001$, Figure R6c).

Figure R6 (i.e., Figure 5 in the main text). **The responses of soil organic carbon (SOC) to organic fertilization across environmental gradients in global grasslands and croplands.** (a) Percent increase in SOC under organic fertilization in grasslands and croplands. (b) Effects of environmental factors on SOC responses under organic fertilization. Circles and error bars represent average parameter estimates (standardized regression coefficients) and 95% confidence intervals in the linear mixed effect models, respectively. (c–h) The brown and blue lines in each panel show model fits using partial regression for each environmental factor, and the brown and blue area around the fit lines represents the 95% confidence intervals. The slopes of the partial regressions are the same as the fixed effects shown in Figure R6b, we compared the difference in slopes between grasslands and croplands using the standardized major axis tests. Equations in c–h show the values of standardized regression coefficients.

Second, we further collected data on quality (C:N ratios) of organic fertilizers,

which were available in about 40% of all the organic studies. The C:N ratios of organic fertilizers were generally similar between grasslands and croplands, ranging from 7 to 72 and from 7 to 80, respectively (Figure R7d). We included C:N ratio in the linear mixed effect models and partial regression analysis. Under organic fertilization, grasslands with higher mean annual temperature, the amount and C:N ratios of organic input, fertilization duration, soil pH, sand content and water content had greater increase in SOC (Figure R7). However, the SOC response ratio to organic fertilization decreased with increasing C:N ratios of organic fertilizer inputs, soil sand content and water content in croplands. The reviewer is correct that the quality of organic input affects its decomposition and retention in soil. Organic fertilizer with higher C:N ratios tended to result in greater SOC increase in grasslands but less so in croplands, suggesting that low quality organic fertilizers may still promote carbon sequestration in grasslands. We have now added this into the Discussion section in the revision (Lines 378-386).

Third, in grasslands, SOC response ratio under organic fertilization increased with increasing mean annual temperature after accounting for the effects of soil properties, the amount and quality of organic fertilizer added (Figure R7). This result is consistent when we used all data (i.e., Figure R6). We included this result in the revision (Lines 259-261).

Figure R7. The responses of soil organic carbon (SOC) to organic fertilization

across environmental gradients in global grasslands and croplands. We used linear mixed effects models to evaluate the impacts of climate, soil properties, the quantity and quality of organic inputs on SOC under organic fertilization. (a) Circles and error bars represent average parameter estimates (standardized regression coefficients) and 95% confidence intervals in the linear mixed effect models, respectively. The environmental factors were obtained from the best model selected based on Akaike Information Criterion (b-i). The brown and blue lines in each panel show model fits using partial regression for each environmental factor, and the brown and blue area around the fit lines represents the 95% confidence intervals. Equations in b-i show the values of standardized regression coefficients. The slopes of the partial regressions are the same as the fixed effects shown in Figure R6 a, we compared the difference in slopes between grasslands and croplands using the standardized major axis tests. **Key:** MAT: mean annual temperature, Org: organic fertilizer amount added, C:N: quality of organic inputs, SOC: initial SOC, Duration: duration year, pH: soil pH, Sand: soil sand content, SWC: soil water content; * $p < 0.05$, ** $p < 0.01$ and *** $p < 0.001$.

Comment #35. Quantity of organic inputs: High organic input can negatively affect annual crop plants (seedlings in particular), but may have limited effects on perennial grasses. Also, it is well-known that low levels of organic C input to croplands may increase crop yields or plant biomass but only have marginal impact on soil organic C. Fig. 5b showed that some organic inputs to grasslands (over 20 g m²) were much higher than those to croplands (the maximum was at 10 g m²). Simply looking at the two regression lines, however, figure 5b seems to suggest that SOC in croplands may be more responsive to high organic fertilization. Then, if similar amounts of organic input were applied, would the C benefit still differ between grasslands and croplands?

Responses: Thank you for your comment. We now explore the differences in carbon storage potential between grasslands and croplands under comparable levels of organic fertilization. (Lines 264-269).

In fact, in this study, we have considered the effects of environmental factors and organic fertilizer rates on SOC by using linear mixed effect model and partial regression analysis (See lines 278-284). After accounting for these background effects, we found that the effects of organic fertilization on SOC increased with organic fertilizer rates in both grasslands and croplands (Figure R6d). Then, we compared the slopes of the regression of SOC response ratio and organic fertilizer rates between grasslands and croplands by using standardized major axis test²³. The results show that there was no significant difference in the increasing rate of SOC when similar amounts of organic fertilizer input were applied to grasslands and croplands ($p=0.304$, Figure R6d).

Comment #36. In Fig. 5c, effect of initial soil C: the difference in the slope of two regression lines (one for croplands and another for grasslands) can be largely attributed to one data point of a high SOC in croplands: that data point has a SOC of ca. 10%, which is very different from most global croplands. Is that fair to draw a general conclusion? I understand it is very challenging to deal with this kind of situation, but it may need to mention this in the Discussion.

Responses: We acknowledge the reviewer's comment. Here, we removed the data point of a high SOC in cropland and re-evaluated the difference in the slope of two regression lines (blue for croplands and brown for grasslands, Figure R8). The results are consistent, the slope of croplands is higher than that of grasslands. Thus, our results were robust. Please see Figure R8.

Figure R8. The responses of soil organic carbon (SOC) to organic fertilization across initial/background SOC in global grasslands and croplands. Blue and brown represent croplands and grasslands, respectively. Panel a and b represent the results before and after deleting a point of high SOC, respectively. The brown and blue lines in each panel show model fits using partial regression for initial/ background SOC.

Comment #37. Quality of organic inputs: Low quality (e.g., high C:N ratio with low contents of available nutrients) of organic fertilizers can have negative effects on crop plants (saying wheat plants) but may still promote growth of some grasses (e.g., switchgrass) because some grasses have lower demands for nutrients. Yet, quality of organic input affects its decomposition and retention in soil.

Responses: Thank you for your insightful comment. Please see our response to Comment #34.

Comment #38. SECOND, the diversity issue: Yes, organic fertilization may maintain or even enhance plant diversity, but this may have different implications for croplands, intensively managed forage fields, and semi-natural grasslands. For a field of crop plants or forage plants, high diversity likely means more weeds, which may compromise both the yield and quality of a crop or forage plants. In contrast, high diversity may be beneficial in natural grasslands. In other words, high diversity may be good for conservation (and other ecosystem services) but does not necessarily mean it is desired in highly managed agroecosystems. May the data further be analyzed to examine the differences among croplands, intensively managed forage fields, and semi-natural grasslands?

Responses: Thanks very much for the comment.

We agree with the reviewer that high biodiversity is not necessarily desired in highly managed agroecosystems, and we did not collect and analyze biodiversity data from agroecosystems (Lines 429-431, in the method section). In highly managed agroecosystems, only one or two crop species are maintained and thus it is not reasonable to analyze biodiversity. Moreover, regardless of whether organic or inorganic fertilization is applied, weeds in agroecosystems are often removed manually or by herbicides. Thus, compared to inorganic fertilization, organic fertilization is not expected to have adverse effect on stability of crop yield and quality.

Based on the reviewer's comment, we have now further clarified the ecosystem in the revision. Inorganic fertilization significantly increased aboveground biomass by 42% ($p < 0.001$) relative to ambient conditions, and caused an 18% ($p < 0.001$) decline in species richness and 6% ($p < 0.001$) decline in evenness (blue triangles, Figures. 1a, b and c) in *semi-natural and natural grasslands* (Lines 155-169, in the result section). Organic fertilization may facilitate restoration of degraded grasslands and abandoned croplands by increasing SOC, plant biomass and biodiversity (Lines 359-360, in the discussion section). Thus, organic fertilizers can serve as a nature climate solution that reduces the tradeoff between increasing aboveground biomass and loss of plant diversity in semi-natural and natural grasslands.

Comment #39. THIRD, general benefits of organic vs. inorganic fertilization: One major message of the manuscript is that organic fertilization may have more benefits in terms of both plant diversity/biomass and soil organic C and provides a climate-friendly option for grassland management. The benefit of organic fertilization is not controversial, and I believe, is also accepted by many farmers and land managers. However, the challenge is that organic fertilization has some problems or disadvantages too. One of them is that the economic cost of transporting and spreading organic

fertilizers is much higher than that for inorganic fertilizers, which cuts into profit margins of farming as an industry. This issue may need to be discussed while recommending organic fertilization.

Responses: Thank you for your helpful suggestion. We have now included some discussion about recommending organic fertilizer in the revision.

Considering the whole-system feedback, increased aboveground biomass production under organic fertilization supports more livestock and provides more manure to fertilize grasslands, and thus further increases grassland biomass production. However, croplands usually receive transported organic fertilizer transported from other locations²⁴, potentially increasing the cost of transporting and spreading, which can reduce profit margins of farming as an industry. See lines 389-391.

Minor edits or typos:

Comment #40. Line 83: mechanisms.

Responses: Agreed and done, as suggested. See line 104.

Comment #41. Figures: the labeling of several figures is confusing. For example, Figure 2 has A and B, as well as a, b, c, d, e, and f. It seems there were some different connections between A (or B) and a, b, c, d, e, and f. May Fig. 2A, Fig.2A1, Fig.2A2 and Fig.2A3 work? Similar situation for Figure 3.

Responses: Here, we refer to this journal format requirements for revision. See Figure 2 and Figure 3, in the results section.

Comment #42. Lines 275-276: Effects of

Responses: Change made accordingly. See line 278.

Comment #43. Lines 370-371: not a complete sentence.

Responses: We have now revised the sentence as follows, “Considering the whole-system feedback, increased aboveground biomass production under organic fertilization supports more livestock and provides more manure to fertilize grasslands, and thus further increases grassland biomass production. However, croplands usually receive imported organic fertilizer from other locations²⁴, potentially increasing the cost of transporting and spreading, and thus reducing profit margins of farming as an industry.” See lines 386-391.

References:

1. Cohen-Shacham, E., Walters, G., Maginnis, S. & Janzen, C. *Nature-based Solutions to address global societal challenges*. (2016).
2. Du, Y. *et al.* Effects of manure fertilizer on crop yield and soil properties in China: A meta-analysis. *Catena* **193**, 104617 (2020).
3. Gautam, A. *et al.* Responses of soil microbial community structure and enzymatic activities to long-term application of mineral fertilizer and beef manure. *Environmental and Sustainability Indicators* **8**, 100073 (2020).
4. Griscom, B. W. *et al.* Natural climate solutions. *Proceedings of the National Academy of Sciences of the United States of America* **114**, 11645–11650 (2017).
5. Seddon, N. *et al.* Understanding the value and limits of nature-based solutions to climate change and other global challenges. *Philosophical Transactions of the Royal Society B-Biological Sciences* **375**, 20190120 (2020).
6. Tadesse, G., Peden, D., Abiye, A. & Wagnew, A. Effect of manure on grazing lands in Ethiopia, East African highlands. *Mountain Research and Development* **23**, 156–160 (2003).
7. Aber, J. *et al.* Nitrogen saturation in temperate forest ecosystems - Hypotheses revisited. *Bioscience* **48**, 921–934 (1998).
8. Tian, Q. *et al.* A novel soil manganese mechanism drives plant species loss with increased nitrogen deposition in a temperate steppe. *Ecology* **97**, 65–74 (2016).
9. Crawley, M. J. *et al.* Determinants of species richness in the park grass experiment. *American Naturalist* **165**, 179–192 (2005).
10. Kidd, J., Manning, P., Simkin, J., Peacock, S. & Stockdale, E. Impacts of 120 years of fertilizer addition on a temperate grassland ecosystem. *Plos One* **12**, e0174632 (2017).
11. Li, B. *et al.* Responses of soil organic carbon stock to animal manure application: A new global synthesis integrating the impacts of agricultural managements and environmental conditions. *Global Change Biology* **27**, 5356–5367 (2021).
12. Rambaut, L.-A. E., Tillard, E., Vayssières, J., Lecomte, P. & Salgado, P. Trade-off between short and long-term effects of mineral, organic or mixed mineral-organic fertilisation on grass yield of tropical permanent grassland. *European Journal of Agronomy* **141** (2022).
13. Ye, J.-S., Bradford, M. A., Dacal, M., Maestre, F. T. & Garcia-Palacios, P. Increasing microbial carbon use efficiency with warming predicts soil heterotrophic respiration globally. *Global Change Biology* **25**, 3354–3364 (2019).
14. Slessarev, E. W. *et al.* Initial soil organic carbon stocks govern changes in soil carbon: Reality or artifact? *Global Change Biology*, 1239–1247 (2022).
15. Kan, Z.-R. *et al.* Mechanisms of soil organic carbon stability and its response to no-till: A global synthesis and perspective. *Global Change Biology* **28**, 693–710 (2022).
16. Liu, Y. *et al.* Meta-analysis on the effects of types and levels of N, P, and K fertilization on organic carbon in cropland soils. *Geoderma* **437**, 116580 (2023).
17. Naeem, M., Ansari, A. & Gill, S. *Contaminants in Agriculture: Sources, Impacts and Management*. (2020).
18. Li, S. X., Wang, Z. H., Hu, T. T., Gao, Y. J. & Stewart, B. A. in *Advances in Agronomy* Vol. 101 (ed Donald L. Sparks) 123–181 (Academic Press, 2009).
19. Butler, T. J. & Muir, J. P. Dairy manure compost improves soil and increases tall wheatgrass yield. *Agronomy Journal* **98**, 1090–1096 (2006).
20. Ryals, R., Eviner, V. T., Stein, C., Suding, K. N. & Silver, W. L. Grassland compost amendments increase plant production without changing plant communities. *Ecosphere* **7**, e01270 (2016).
21. Cai, A. *et al.* Manure acts as a better fertilizer for increasing crop yields than synthetic fertilizer does by improving soil fertility. *Soil and Tillage Research* **189**, 168–175 (2019).
22. De Melo, T. R. *et al.* Biogenic aggregation intensifies soil improvement caused by manures. *Soil and Tillage Research* **190**, 186–193 (2019).
23. Warton, D. I., Wright, I. J., Falster, D. S. & Westoby, M. Bivariate line-fitting methods for allometry. *Biological Reviews* **81**, 259–291 (2006).
24. Gaudare, U. *et al.* Soil organic carbon stocks potentially at risk of decline with organic farming expansion. *Nature Climate Change* **13**, 719 – 725 (2023).

Reviewers' Comments:

Reviewer #1:

Remarks to the Author:

Under a second review is a global meta-analysis comparing organic and inorganic fertilizer effects on plant biomass, plant diversity, and soil carbon in grasslands. As I mentioned before, I appreciate the efforts and results of this study. There are some neat findings in the study and I appreciate the improvements the authors made (especially with the figures).

Unfortunately, the authors did not make substantial improvements in the writing. The combination of poor writing and massive amount of findings are difficult to sort out a clear and coherent story. I recommend either another round of major revisions or rejection if they cannot make major improvements in writing after this 2nd review. Perhaps this paper is better suited to journal that has a longer format, that would allow the authors can delve further into the details and clarify the story?

GENERAL COMMENTS:

1. Writing could still use a good amount of work – the authors did not polish the writing as I originally recommended. There are sections/sentences that remain too convoluted and difficult to read. Avoid “Higher” in most cases and use other word for greater or more of something. There were 16 uses of the word “higher”. Most of them are probably not appropriate. E.g., see line 319. The phrase “higher soil moisture support higher plant species richness” just does not sound right and is redundant. There are many errors and awkward phrasings still left in this manuscript. I highlight a few of the most egregious in the specific comments. The authors need to comb through the manuscript carefully and improve the writing.

2. There is too much going on and not enough space and clarity to explain it all. I think part of the issue is the convoluted findings with cropland and grassland. More space is needed to tease apart the whole land-use interaction. I thought if the authors were to improve the writing, then a clearer story would emerge. Unfortunately, the writing wasn't sufficiently improved, and the story may have gotten even murkier. There are entire meta-analyses just on N addition in croplands for example. Perhaps Nature Communications is not the appropriate journal? See my suggestion about considering submitting to a different journal.

3.

SPECIFIC COMMENTS:

L30. This level of speculation, which is also a bit of a tangent, is not necessary in an Abstract.

L49. Why is ammonium nitrate included here? It seems like a distraction. I'm sure there were many other forms of fertilizer added and the audience doesn't need to know that.

L80-81. Replace “contribute to the enhancement of soil organic carbon” with “increase soil organic carbon”. Simple wording can make the sentence much clearer.

L90. Replace “reduction in” with “reduced”

L95-97. This sentence is poorly worded and too long. Consider using something more similar to “Organic fertilizers added to grasslands increase carbon storage without a tradeoff in plant diversity loss.”

L110. “Balanced” is not the appropriate word here. Maybe rephrase this more precisely and clearly. I think the authors are trying to say that there are other macro/micronutrients added in addition.

L113-116. This is a lot of specific examples. Can't the authors just cite the other meta-analyses on manure application and SOC change (e.g., Maillard & Angers 2014; and other more recent ones).

L144-146. This sentence and logic is confusing. Suggest a major re-write.

L195. This is perfect example where the writing still really needs to be improved.

Fertilizer/fertilization is unnecessarily used twice. Instead of this hard-to-read, awkward sentence the authors could have simply wrote, "Increasing inorganic fertilizer rate in grasslands reduced species richness".

L186. Remove "(i.e., In RR)".

L187. Can probably use "Greater" rather than "higher" here.

L192. Replace "grassland sites" with "grasslands"

L200. Species richness is not a proxy but is a measure of biodiversity. This doesn't really seem necessary to state in a figure caption anyways. Same with L205-206. This can go in Materials & Methods.

L226-228. Difficult to read and awkwardly worded sentence. Why not just write, "Inorganic and organic fertilizers equally increased aboveground biomass but the former also decreased diversity."

L232-238. This is a lot of words (space) describing something in supplemental. Either get rid of the words or move the figure to main manuscript.

L257-259. Rewrite sentence. It could help to replace "amount added" with "rate".

L292-395. The Discussion. Most of this is too convoluted and reads more like a Results section. I suggest re-writing for clarity and synthesize most important findings.

L319. See General Comment #1.

L345. Change "fertilizer releases slowly nutrients" to "fertilizers release nutrients slowly"

L348. Don't forget P and S too! I mentioned K, Ca, and Mg in the last review – I didn't mean for the authors to take me literally. But maybe just phrase it as "macro- and micro-nutrients".

L368-370. For which fertilizer source?

L371-374. This section can be written much clearer.

L380. Use "warmer" rather than "higher"

L384. Use "wider" rather than "higher"

L405-407. But is it really soil moisture or the other way around? Maybe the effect of the fertilizer on biodiversity then affects soil moisture. You can imagine that different species alter microclimate, rooting patterns, and transpiration. This idea that the soil moisture is regulating this relationship is not founded in solid evidence. Change it in the Discussion too.

Figure 1. Remove the illustrations and text summarizing the figures. I don't think this makes the data any clearer and is just distracting. I don't know what the blue lines are in the soil. Are the mycorrhizae or water? Are there rocks in the soil too?

Reviewer #2:

Remarks to the Author:

This revision has addressed my concerns on the previous version. I now feel comfortable for this manuscript to be accepted for publication. Congrats to the authors for this comprehensive and

effective synthesis.

Response to Reviewers

Reviewer #1 (Remarks to the Author):

Under a second review is a global meta-analysis comparing organic and inorganic fertilizer effects on plant biomass, plant diversity, and soil carbon in grasslands. As I mentioned before, I appreciate the efforts and results of this study. There are some neat findings in the study and I appreciate the improvements the authors made (especially with the figures).

Unfortunately, the authors did not make substantial improvements in the writing. The combination of poor writing and massive amount of findings are difficult to sort out a clear and coherent story. I recommend either another round of major revisions or rejection if they cannot make major improvements in writing after this 2nd review. Perhaps this paper is better suited to journal that has a longer format, that would allow the authors can delve further into the details and clarify the story?

Responses: Thanks very much for the comments and we are sorry to learn that you found the writing unacceptable. We have once again substantially rewritten many sections of the text to improve clarity. Please see the point-by-point response below.

GENERAL COMMENTS:

Comment #1. Writing could still use a good amount of work – the authors did not polish the writing as I originally recommended. There are sections/sentences that remain too convoluted and difficult to read. Avoid “Higher” in most cases and use other word for greater or more of something. There were 16 uses of the word “higher”. Most of them are probably not appropriate. E.g., see line 319. The phrase “higher soil moisture support higher plant species richness” just does not sound right and is redundant. There are many errors and awkward phrasings still left in this manuscript. I highlight a few of the most egregious in the specific comments. The authors need to comb through the manuscript carefully and improve the writing.

Responses: We have replaced “higher” with other words as recommended by the reviewer. We have also reduced the redundancy in writing, and simplified phrasing throughout the manuscript.

Comment #2. There is too much going on and not enough space and clarity to explain it all. I think part of the issue is the convoluted findings with cropland and grassland. More space is needed to tease apart the whole land-use interaction. I thought if the authors were to improve the writing, then a clearer story would emerge. Unfortunately, the writing wasn't sufficiently improved, and the story may have gotten even murkier. There are entire meta-analyses just on N addition in croplands for example. Perhaps

Nature Communications is not the appropriate journal? See my suggestion about considering submitting to a different journal.

Responses: We have substantially improved the link between grasslands and croplands in the Introduction, Results and Discussion sections. We have also clarified the difference and similarity between croplands and grasslands following organic fertilization (L309-323). After a careful revision and editing, we now believe the story is clear and holistic.

SPECIFIC COMMENTS:

Comment #3. L30. This level of speculation, which is also a bit of a tangent, is not necessary in an Abstract.

Responses: Agreed, we removed the sentence.

Comment #4. L49. Why is ammonium nitrate included here? It seems like a distraction. I'm sure there were many other forms of fertilizer added and the audience doesn't need to know that.

Responses: Thanks very much for the suggestion, we have now removed ammonium nitrate here. See line 45.

Comment #5. L80-81. Replace “contribute to the enhancement of soil organic carbon” with “increase soil organic carbon”. Simple wording can make the sentence much clearer.

Responses: Agreed and done. See line 77.

Comment #6. L90. Replace “reduction in” with “reduced”

Responses: Change made. See line 86.

Comment #7. L95-97. This sentence is poorly worded and too long. Consider using something more similar to “Organic fertilizers added to grasslands increase carbon storage without a tradeoff in plant diversity loss.”

Responses: Thanks very much for the suggestion, change made accordingly. See lines 89-90.

Comment #8. L110. “Balanced” is not the appropriate word here. Maybe rephrase this more precisely and clearly. I think the authors are trying to say that there are other macro/micronutrients added in addition.

Responses: Thanks very much for the suggestion, we have now revised the sentence

and removed “Balanced”. See lines 75-78.

Comment #9. L113-116. This is a lot of specific examples. Can't the authors just cite the other meta-analyses on manure application and SOC change (e.g., Maillard & Angers 2014; and other more recent ones).

Responses: Agreed and done. See lines 100-102.

Comment #10. L144-146. This sentence and logic is confusing. Suggest a major re-write.

Responses: We have now revised the sentence to read: if nitrogen detriment (e.g., acidification) was the main mechanism, organic fertilization would not cause biodiversity loss in grasslands. See lines 123-125.

Comment #11. L186. Remove “(i.e., ln RR)” .

Responses: Change made. See line 160.

Comment #12. L187. Can probably use “Greater” rather than “higher” here.

Responses: Change made. See line 161.

Comment #13. L192. Replace “grassland sites” with “grasslands”

Responses: Change made. See line 163.

Comment #14. L195. This is perfect example where the writing still really needs to be improved. Fertilizer/fertilization is unnecessarily used twice. Instead of this hard-to-read, awkward sentence the authors could have simply wrote, “Increasing inorganic fertilizer rate in grasslands reduced species richness” .

Responses: Change made. See lines 168-169.

Comment #15. L200. Species richness is not a proxy but is a measure of biodiversity. This doesn't really seem necessary to state in a figure caption anyways. Same with L205-206. This can go in Materials & Methods.

Responses: Thanks very much for the suggestion, we have now revised and moved this information to the Methods section (L432-434).

Comment #16. L226-228. Difficult to read and awkwardly worded sentence. Why not just write, “Inorganic and organic fertilizers equally increased aboveground biomass but the former also decreased diversity.”

Responses: Thanks very much for the suggestion, change made. See lines 193-195.

Comment #17. L232-238. This is a lot of words (space) describing something in supplemental. Either get rid of the words or move the figure to main manuscript.

Responses: Thanks very much for the suggestion, we have removed these words.

Comment #18. L257-259. Rewrite sentence. It could help to replace “amount added” with “rate” .

Responses: We have revised the sentence to read: After accounting for soil properties and organic fertilizer rate, the effect of organic fertilization on SOC increased significantly with mean annual temperature in grasslands. (L215-217).

Comment #19. L292-395. The Discussion. Most of this is too convoluted and reads more like a Results section. I suggest re-writing for clarity and synthesize most important findings.

Responses: We have carefully revised the Discussion, which we think focuses on the most important findings to highlight. Now the Discussion section repeats key results and then relates those results to other studies or potential management recommendations.

Comment #20. L319. See General Comment #1.

Responses: We have now revised the sentence to read: Previous studies also supported this finding that mesic grasslands had greater plant species richness than dry grasslands (L260-261).

Comment #21. L345. Change “fertilizer releases slowly nutrients” to “fertilizers release nutrients slowly”

Responses: Change made. See line 285.

Comment #22. L348. Don’ t forget P and S too! I mentioned K, Ca, and Mg in the last review – I didn’ t mean for the authors to take me literally. But maybe just phrase it as “macro- and micro-nutrients” .

Responses: Thanks very much for the suggestion. We have now revised the sentence based on Reviewer’s comment, and expanded the discussion according to Editor’s suggestion.

Therefore, organic fertilizer releases nutrients slowly as it decomposes and may continuously provide multiple macro- and micro-nutrients for plant growth¹⁻³. Based

on a subset (about 6% of all studies) of the data that measured both macro- and micro-nutrients (Table S14), organic fertilizers supply nitrogen, phosphorus, potassium, magnesium, calcium, zinc, iron, etc. As a result, organic fertilizers increased more biomass than did inorganic fertilizers ($56\% \pm 5\%$ versus $42\% \pm 2\%$; Fig.1). Previous study proposed that the average supply rate of the most limiting resource controlled plant species diversity⁴. Organic fertilizers provide multiple nutrient resources and thus may avoid species loss. Collectively, organic fertilization increases soil fertility and aboveground biomass while limiting detrimental effects on plants and microbes. (L 284-294).

Comment #23. L368-370. For which fertilizer source?

Responses: We have revised the sentence to read: We found that SOC response ratio under organic fertilization increased in grasslands but decreased in croplands under warmer mean annual temperature (Fig. 5c). (L309-310).

Comment #24. L371-374. This section can be written much clearer.

Responses: We have now improved the section: This difference may result from management practices, where croplands are typically tilled while grasslands are not. Tillage can disrupt soil aggregates and accelerate SOC decomposition by microorganisms (L311-313).

Comment #25. L380. Use “warmer” rather than “higher”

Responses: Change made. See line 320.

Comment #26. L384. Use “wider” rather than “higher”

Responses: Change made. See line 331.

Comment #27. L405-407. But is it really soil moisture or the other way around? Maybe the effect of the fertilizer on biodiversity then affects soil moisture. You can imagine that different species alter microclimate, rooting patterns, and transpiration. This idea that the soil moisture is regulating this relationship is not founded in solid evidence. Change it in the Discussion too.

Responses: We have now changed the sentence: Organic fertilization increased plant species in grasslands with greater soil moisture (L344-345).

We also changed it in the Discussion (L250-262).

Comment #28. Figure 1. Remove the illustrations and text summarizing the figures. I don't think this makes the data any clearer and is just distracting. I don't know what the blue lines are in the soil. Are the mycorrhizae or water? Are there rocks in the soil too?

Responses: According to the reviewer's suggestion, we simplified the illustrations and removed the blue lines and rocks. The illustrations and text summarized the magnitudes of changes in plant aboveground biomass, biodiversity and soil organic carbon following organic and inorganic fertilization, which might be helpful to readers outside the field of specialty. We hope the reviewer agree.

Reviewer #2 (Remarks to the Author):

Comment #1. This revision has addressed my concerns on the previous version. I now feel comfortable for this manuscript to be accepted for publication. Congrats to the authors for this comprehensive and effective synthesis.

Responses: Thanks very much for the comments and suggestion in the last review, which greatly improved the manuscript and synthesis of results.

References:

- Rambaut, L.-A. E., Tillard, E., Vayssieres, J., Lecomte, P. & Salgado, P. Trade-off between short and long-term effects of mineral, organic or mixed mineral-organic fertilisation on grass yield of tropical permanent grassland. *European Journal of Agronomy* **141**, (2022).
- Li, B. *et al.* Responses of soil organic carbon stock to animal manure application: A new global synthesis integrating the impacts of agricultural managements and environmental conditions. *Global Change Biology* **27**, 5356–5367, (2021).
- Kidd, J., Manning, P., Simkin, J., Peacock, S. & Stockdale, E. Impacts of 120 years of fertilizer addition on a temperate grassland ecosystem. *Plos One* **12**, e0174632, (2017).
- Stevens, M. H. H. & Carson, W. P. Resource quantity, not resource heterogeneity, maintains plant diversity. *Ecology Letters* **5**, 420–426, (2002).